# California Wildfire Smoke Contributes to a Positive Temperature Anomaly over the Western United States

James L. Gomez[1], Robert J. Allen[1], and King-Fai Li[2]

[1]Department of Earth & Planetary Sciences, University of California Riverside, Riverside, USA
[2]Department of Environmental Sciences, University of California Riverside, Riverside, USA

**Correspondence:** James Gomez (jgome222@ucr.edu)

**Abstract.** Wildfires in the southwestern United States, particularly in northern California (nCA), have grown in size and severity in the past decade. As they have grown larger, they have been associated with large emissions of absorbing aerosols and heat into the troposphere. Utilizing satellite observations from MODIS, CERES, and AIRS, as well as reanalysis from MERRA-2, the meteorology associated with fires during the wildfire season (June-October) were discerned over the nCA-NV (northern California and Nevada) region in the 2003-2022 time period. Wildfires in the region have a higher probability of occurring on days of positive temperature $T$ anomalies and negative relative humidity $RH$ anomalies, making it difficult to discern the radiative effects of aerosols that are concurrent with fires. To attempt to better isolate the effects of large fire emissions on meteorological variables, such as clouds and precipitation, variable anomalies on high fire emission days (90th percentile) were compared to low fire emission days (10th percentile) and were further stratified based on whether surface relative humidity $RH_s$ was anomalously high (75th percentile) or low (25th percentile) compared to typical fire season conditions. Comparing the high fire emission/high $RH_s$ data to the low fire emission/high $RH_s$ data, positive tropospheric $T$ anomalies were found to be concurrent with positive $AOD$ anomalies. Further investigation found that due to shortwave absorption, the aerosols heat the atmosphere at a rate of 0.04-0.09 Kday$^{-1}$, depending on whether $RH$ conditions are anomalously positive or negative. The positive $T$ anomalies were associated with significant negative 850 hPa-300 hPa $RH$ anomalies during both 75th percentile $RH_s$ conditions. Furthermore, high fire emission days under high $RH_s$ conditions are associated with negative $CF$ anomalies that are concurrent with the negative $RH$ anomalies. This negative $CF$ anomaly is associated with a significantly negative regional precipitation anomaly and an positive net top of atmosphere radiative flux anomaly (a warming effect) in certain areas. The $T$, $RH$, and $CF$ anomalies under the high fire emission/high $RH_s$ conditions compared to low fire/high $RH_s$ conditions correlate significantly spatially with $AOD$ anomalies. Additionally, the vertical profile of these variables under the same stratification are consistent with positive black carbon mass mixing ratio anomalies from MERRA-2. However, causality is difficult to discern, and further study is warranted to determine to what extent the aerosols are contributing to these anomalies.

## 1 Introduction

As a result of climate change, land use change, and forest management, the frequency of wildfires in California has trended upward from 100 fires per year in the 1920s to 300 fires per year in the late 2010s (Li & Banerjee, 2021). The size of these

wildfires has also increased, with total burned area (square distance burned by a fire) increasing from roughly 1000 km$^2$ to almost 4000 km$^2$ in the same time period (Li & Banerjee, 2021). According to a recent study, frequency of extreme daily wildfire events in the region are projected to increase by 59%-172% in coming years due to climate change (Brown et al., 2023), which is consistent with findings of numerous other studies (Palinkas, 2020; Ager et al., 2021; United Nations Environment Programme, 2022). In both higher and lower $CO_2$ mitigation scenarios, large wildfire events are projected to become

more commonplace by the end of the 21st century worldwide, as well in the southwestern US (United Nations Environment Programme, 2022). Large wildfire events in the late 2010's and early 2020's were associated with more intense "fire weather": high temperatures $T$, low relative humidity $RH$, and high surface wind speeds $U_s$ (Varga et al., 2022; Keeley & Syphard, 2019). These fire weather conditions may be potentially intensified, or alleviated, by the fires themselves. Higher burn severity wildfires, such as the 2020 wildfires in California (CA), have been observed to inject smoke plumes higher into the troposphere

than in previous years (Wilmot et al., 2022). These smoke plumes consist of both shortwave (SW) absorbing aerosols such black carbon $BC$ and reflective aerosols such as organic aerosol (OA), as well as brown carbon, which is absorbing and reflective. Additionally, wildfires have also been associated with emission of other aerosol species through feedbacks. While dust is not emitted from biomass burning, a number of studies have linked fires to concurrent dust emission through creation of convective updrafts (Wagner et al., 2018, 2021) and delayed dust emissions through wildfire clearing of vegetation (Wagenbrenner

et al., 2013, 2017; Yu & Ginoux, 2022). The absorbing properties of wildfire smoke and co-emitted dust over the western US, measured using absorbing aerosol optical depth (AAOD), is uncertain. However, a recent study of CA fires indicates that wildfires increase AAOD relative to the annual mean by tenfold (Cho et al., 2022). An injection of absorbing aerosols into the troposphere may cause a local warming affect, altering the hydrological and radiative balance of the atmosphere (Allen & Sherwood, 2010; Thornhill et al., 2018; Allen et al., 2019; Herbert & Stier, 2023). Smoke plumes that reach the upper troposphere

(pressures<500 hPa) may deposit absorbing aerosols that could burn off high clouds, and promote more stable low clouds (Stjern et al., 2017; Smith et al., 2018; Allen et al., 2019), leading to SW and longwave (LW) cooling. Alternatively, if the absorbing aerosols are concurrent with low clouds, the relative humidity of the liquid cloud layer would be decreased, burning off low clouds and leading to a decrease in outgoing SW flux (Koch & Del Genio, 2010; Allen & Sherwood, 2010). These are both examples of aerosol semi-direct effects. Past observations and modelling experiments have shown dust aerosol is

associated with semi-direct effects (Tsikerdekis et al., 2019; Amiri-Farahani et al., 2017; Helmert et al., 2007), as dust also has SW absorbing properties (Highwood & Ryder, 2014; Kok et al., 2023). Furthermore, the higher altitude of absorbing aerosol from California fires may alter cloud microphysics, which also has the potential to change the radiative balance of the surface and atmosphere. An influx of aerosols into the troposphere may create an abundance of cloud condensation nuclei (CCN) for droplets to condense onto, decreasing effective radius $R_{eff}$ of the clouds, an effect already observed with smoke (OA and

$BC$) particles in the northwestern US (Twohy et al., 2021). A decrease in $R_{eff}$ would increase the albedo of the clouds, assuming constant water path, which would then increase outgoing SW radiation. This decrease in $R_{eff}$ can also affect liquid water path $LWP$ as the smaller droplets can evaporate much faster than larger droplets, or the smaller droplets can suppress precipitation, which increases $LWP$ by reducing the liquid water leaving the cloud (Goren & Rosenfeld, 2012). The lighter droplets can also be lofted higher in the atmosphere, where they condensate further and release latent heat, then eventually fall

from this greater height and evaporate. Therefore, to compensate, polluted clouds have more intense updrafts and downdrafts than pristine clouds (Khain, 2009). SW absorption itself can also decrease precipitation $P$ in other ways, such as reducing SW radiation reaching the surface or through rapid atmospheric adjustments (Sand et al., 2020; Samset, 2022; Allen et al., 2023).

Large Fires are not only limited to the western US. Australia, the Mediterranean Basin, and South America have all experienced an increase in large fire events due to climate change and land management (Shi et al., 2021; Ruffault et al., 2020; Artaxo et al., 2013; Allen et al., 2024a). As the western US, and other parts of the world, enter this new regime of large fires, there comes a need for improved understanding of the effects of aerosols emitted primarily (through biomass burning), secondarily (oxidation of emitted volatiles), or through feedbacks (such as dust emissions concurrent with fires) by wildfires. Models participating in the Coupled Model Intercomparison Project version 6 (CMIP6) (Eyring et al., 2016) do not have parametrizations of biomass burning (BB) aerosol emissions that respond to $CO_2$ emissions in most of their experiments, including the DECK (Diagnosis, Evaluation, and Characterization of Klima) experiments (Gomez et al., 2023). The models that have interactive BB aerosol emissions tend to parameterize them as a function of fuel flammability (temperature and moisture), fuel density, and plant functional type (Mangeon et al., 2016; Li et al., 2019). Most models participating in CMIP6 do not have dynamic vegetation models (Li et al., 2019), and therefore are incapable of incorporating fire-dust feedbacks. Instead, modelers rely on prescription of BB aerosols in most experiments.

Recent modelling experiments have found significant effects of wildfires on regional and global climate scales. Previously, using prescribed aerosol simulations in the Community Earth System Model version 2 (CESM2), it was shown that the large 2019 wildfires in Australia could have intensified that year's La Niña through aerosols directly cooling the ocean surface (Fasullo et al., 2021). Another CMIP6 study observed a similar effect on La Niña as a result of a teleconnection caused by an influx of absorbing aerosols into the atmosphere from South African wildfires (Amiri-Farahani et al., 2020). Biomass burning aerosols may also have other effects on large scale ocean circulation, such as an invigoration of the Atlantic Meridional Overturning Circulation (Allen et al., 2024b). As far as the southwestern US is concerned, a modeling experiment using the WRF/CHEM model was run to analyze the effects of a wildfire event on weather forecasts (Chen et al., 2014). This study found that the BB aerosols suppressed convection, prevented cloud formation, and decreased precipitation. While studies such as these demonstrate that it is possible to model past effects of fires on local and global climate, without parameterization of BB aerosol emission, as well as parametrization of secondary dust aerosol emission from wildfire-cleared vegetation, the radiative forcing of future fires' primary and secondary aerosols will remain a source of uncertainty. Furthermore, there are few, if any studies, that attempt to discern the impacts of large fires over the southwestern US. Twohy et al. (2021) analyzed satellite observations of cloud microphysical properties over part of the region, however this study was conducted during only one wildfire event in 2018. As a result, there is no comprehensive long-term observational study over the southwestern US concerning wildfire aerosol-cloud interaction. Therefore, to further understand the effects of wildfires on the climate of one of the most populated areas in the US, this paper aim to identify radiative as well as microphysical effects that these aerosols may have in the region under different atmospheric conditions utilizing satellite data.

## 2   Satellite and Reanalysis Datasets

The objective of this study is to quantify the impacts of wildfire aerosol emissions on meteorological parameters, such as clouds and precipitation, over the southwestern US using observations. This includes the Aqua satellite with the MODIS, AIRS and CERES instruments. The Modern-Era Retrospective analysis for Research and Applications, Version 2 (MERRA-2) reanalysis project (Randles et al., 2017; Global Modeling And Assimilation Office & Pawson, 2015) is used to obtain daily black carbon mass mixing ratio vertical profiles. Fire dry matter emission data $DM$ are used as a proxy for fire severity, and are derived from the Global Fire Emissions Database version 4 (GFED4) (van der Werf et al., 2017; Randerson et al., 2017). All data sets are globally gridded observational data sets, with the exception of GFED4 and MERRA-2 which are considered globally gridded reanalysis datasets.

### 2.1   Global Fire Emissions Database (GFED4)

GFED4 $DM$ emissions are calculated in the Carnegie–Ames–Stanford Approach (CASA) model, which requires MODIS burned area data, meteorological data from the ERA-Interim reanalysis dataset, photosynthetically active radiation data based on Advanced Very high-resolution Radiometer satellite instrument retrievals, and vegetation continuous fields data from the MODIS MOD44B dataset (van der Werf et al., 2017). $DM$ is the emission of any gas or aerosol from burned vegetation, and a list of all these types of emission can be found in van der Werf et al. (2017). The CASA model is run using burned area data from combined MODIS-Aqua and MODIS-Terra level 3 data (MCD64A1). Wildfire studies tend to either use fire power (from MODIS or VIIRS) or burned area-based datasets to quantify fire severity. Burned area is determined by MODIS from a time series of burn sensitive vegetation index, which compares daily surface reflectances (Giglio et al., 2018). Fire power is the radiated energy from fires over time, and MODIS determines this quantity by comparing the brightness temperature of a fire pixel to the background brightness temperature (Peterson et al., 2013). Use of a burned area-based dataset is preferable to a fire power dataset for this paper, as cloud cover may obstruct fire power data retrievals, leading to an underestimation of fire size/severity in a given time period. While cloud cover can also block burned area retrievals, burned area can be recorded once cloud cover has been dissipated, unlike fire power. This introduces a temporal uncertainty, however. This temporal uncertainty is $\pm1$ day for clear sky conditions, $\pm5$ days under consistent 75% cloud cover, and up to $\pm20$ days over persistently very cloudy (85% or higher) intervals (Giglio et al., 2013). However, this temporal uncertainty is likely of little significance for this paper, as cloud cover over the western US during the wildfire season is rarely persistently high (aside from "June gloom" in coastal regions), and the lifetime of biomass burning aerosols (roughly 4-12 days) is generally greater than or equal to the temporal uncertainty of clear sky or consistently cloudy burned area data (Cape et al., 2012). The daily underestimation of fire power is demonstrated in **Figure S1**, which indicates that Aqua fire power retrievals, taken from the MYD14A1 dataset (Giglio & Justice, 2015), underestimate daily fire severity compared to $DM$ with 98% of days reporting a lower normalized fire power than normalized $DM$. Therefore, for fire power to be a more useful metric, a daily combined Aqua/Terra/VIIRS dataset would have to be used, which is not available for the time period of interest. GFED4 fire emissions are also preferred over fire power data and raw burned area data as calculation of fire emissions takes vegetation type and net primary production into account.

Raw burned area and fire power datasets yield information about fire size and intensity, but as aerosol emission also depends on the type of vegetation being burned, use of either dataset over a fire emissions dataset may under-estimate or over-estimate biomass burning aerosol impacts on clouds. However, use of GFED4 data has drawbacks. While use of burned area data reduces the chance of an underestimation of fire impacts, the previously mentioned temporal uncertainty is introduced. Additionally, the CASA model itself is associated with uncertainties. Calculation of net primary production in the model, for example, does not take meteorological variables into account (Liu et al., 2018). As a result, caution must be taken when analyzing the results. To ensure results are robust, the GFED4 $DM$ stratification method was verified by analyzing MODIS $AOD$ anomalies (see Section 2.2) during large fire events (Section 3.3, Section 4.2), and by performing cross correlations between $AOD$ and $DM$ (Supplement section 1, **Figure S2**). GFED4 emissions and burned area data are available from 1997-2016. Data for 2017-2022 is also available, but the data is in "beta" and therefore is more limited. Both the complete and the beta data contain total carbon emissions, as well as dry matter emission. GFED4 also estimates the contribution of 6 different types of vegetation biomes (boreal forest, temperate forest, grassland, agriculture, and peat) to the carbon and dry matter emissions. However, the beta dataset only estimates these contributions for $DM$. Therefore, $DM$ is used as a proxy for the severity of a given fire's emissions, as it is the only variable that both the complete and beta data contain and speciate. All observational datasets utilized in this study have a $1^o$ resolution, however GFED4 emission data is of a $0.25^o$ resolution. Therefore, this data was regridded to a $1^o$ grid. It should be noted that GFED5 has recently been released (Chen et al., 2023), however this dataset was not used as it does not yet include emissions, only has data available up to 2020, and was released after analysis for this paper had concluded.

## 2.2 Aqua

*MODIS-Aqua*: Cloud and aerosol optical depth ($AOD$) data were derived from Moderate Resolution Imaging Spectroradiometer (MODIS) level 3 data. Specifically, the MODIS collection 6.1 $1^o$ level 3 product (MYD08_D3) (Platnick et al., 2003; Salomonson et al., 2002; MODIS Atmosphere Science Team, 2017) is utilized, which yields daily retrieval products from the Aqua satellite. The Aqua satellite makes two overpasses for the region of interest: one ascending run from 2-3 PM, and one descending run from 2-3 AM. The descending dataset is used as most MODIS level 3 cloud property products provided are descending (morning) only. For MODIS cloud retrievals during periods of large $AOD$, especially when the aerosols are concurrent with clouds, it is possible for MODIS to misidentify aerosols as clouds (Herbert & Stier, 2023). This may cause errors in cloud property retrievals, as well as an overestimation of cloud fraction $CF$. This may lead to overestimation of $CF$ during anomalously large fire events. While the MODIS Dark-Target and Deep Blue $AOD$ algorithms are extensively quality controlled and evaluated (Levy et al., 2013; Platnick et al., 2017; Wei et al., 2019), there is still room for errors in $AOD$ and cloud retrieval. Additionally, as it is not possible to distinguish wildfire $AOD$ from other $AOD$, whenever possible, fire emissions from GFED4 are used to discern the impacts of fires on cloud properties.

*AIRS*: Data concerning $T$, water mass mixing ratio $M_{H_2O}$, $CF$, and $RH$ profiles, as well as surface temperature $T_s$ and surface relative humidity $RH_s$, were derived from Atmospheric Infrared Sounder (AIRS) level 3 daily data (AIRS3STD) (AIRS

Science Team & Texeira, 2013). As with the MODIS data, the descending dataset is used.

*CERES*: Top of atmosphere as well as in-atmosphere radiative flux data was derived from Clouds and the Earth's Radiant Energy System (CERES) level 3 daily 1 degree Synoptic product (SYN1deg-Day) (Doelling, 2016, 2017, 2023). This is a combined Terra and Aqua dataset from 2002-2021, and for 2022 it is a combined Terra and NOAA-20 dataset. This CERES dataset combines cloud data from MODIS/VIIRS, aerosol data from GEOS, and top of atmosphere radiative flux data from CERES to produce all-sky, clear-sky, and aerosol-free radiative flux profiles.

## 2.3 GPCP Combined Precipitation Dataset

$P$ data for this project was derived from the daily Global Precipitation Climatology Project (GPCP daily) Climate Data Record, Version 1.3 dataset (Huffman et al., 2001; Adler et al., 2018). GPCP combines satellite observations as well as rain gauge data to produce $1^o$ daily precipitation amount data.

## 2.4 MERRA-2 Aerosol Profiles

Daily vertical black carbon aerosol mass mixing ratio profiles are derived from the M2I3NVAER data product (Global Modeling And Assimilation Office & Pawson, 2015; Buchard et al., 2015). This product estimates aerosol profiles by assimilating MODIS $AOD$ into the GEOS5 model, which is radiatively coupled to the Goddard Chemistry, Aerosol, Radiation, and Transport (GOCART) aerosol module. The GOCART model includes biomass burning emissions from the NASA Quick Fire Emission Dataset (QFED) version 2.1, which provides daily biomass burning aerosol estimates (Buchard et al., 2015). These profiles were then validated using ground and satellite observations of aerosol profiles. This dataset has been previously used to determine effects of wildfire aerosols in other parts of the world (Raga et al., 2022; Nguyen et al., 2020). The aerosol profiles are archived in a high-resolution hybrid sigma pressure grid, and therefore must be interpolated into 1 degree grid cells, and converted into traditional pressure levels. For the purposes of this paper, only the black carbon variables are analyzed. MERRA-2 separates $BC$ into two types: hydrophobic black carbon $BC_{pho}$ and hydrophilic black carbon $BC_{phi}$.

## 2.5 CALIPSO

The Cloud-Aerosol Lidar and Infrared Pathfinder Satellite Observation satellite (CALIPSO) provides observations of aerosol extinction coefficient profiles. MERRA-2 profiles are utilized in the main analysis instead of CALIPSO profiles as gridded CALIPSO data is too low resolution, and is monthly as opposed to daily. Additionally, CALIPSO started collecting data in 2006, which makes the satellite not temporally consistent with MODIS and AIRS, which started collecting data in 2002. More information on CALIPSO can be found in Supplement Section 2.

# 3 Methods

 ## 3.1 Statistics

The bulk of the analysis for this paper involves empirical cumulative distribution functions (CDFs). Empirical distribution functions are calculated for each variable of interest under differing fire and meteorological conditions, and the shift in each distribution is compared. Plotting two CDFs on the same axis allows for comparison on how likely an anomaly is to be positive or negative under differing circumstances, such as how likely a positive/negative anomaly for a certain variable is to occur during a high (90th percentile) fire dry matter emission ($DM90$) or low (10th percentile) fire dry matter emission ($DM10$) event. The 90th percentile is chosen as the purpose of this paper is to analyze the effects of large fire events on climate, not the effects of fires in general. From the calculated normal distributions, the effect size of one variable's distribution on another variable's distribution are estimated using Cohen's d $d$. $d$ is an approximation of by how many standard deviations $\sigma$s the distribution shifts in response to a change in a variable. In this paper, $d$ is calculated to determine the effect size of $DM$ on other variables. $d$ is approximated using

$$d = \frac{\bar{a} - \bar{b}}{0.5\sqrt{\sigma_a^2 + \sigma_b^2}} \tag{1}$$

where $\bar{a}$ is the mean of the ($DM90$) group (group $a$), and $\bar{b}$ is the mean of the ($DM10$) group (group $b$), $\sigma_a$ is the standard deviation of group $a$, and $\sigma_b$ is the standard deviation of group $b$. $d$=0.2-0.5 is considered to be a weak effect, $d$=0.5-0.8 is a moderate effect, and $d$=0.8 or higher is classified as a strong effect.

When comparing two data sets, a two-tailed pooled t-test is used to assess significance, where the null hypothesis of a zero difference is evaluated, with $n_1$+$n_2$-2 degrees of freedom, where $n_1$ and $n_2$ are the number of elements in each data set respectively. Here, the pooled variance

$$s^2 = \frac{(n_1 - 1)S_1^2 + (n_2 - 1)S_2^2}{n_1 + n_2 - 2} \tag{2}$$

is used, where $S_1$ and $S_2$ are the sample variances. For the purposes of this project, the t-test is evaluated at 90% significance.

## 3.2 Data Stratification and Comparison

In section 3.1, it was mentioned that CDFs for variable anomalies during anomalously high and low $DM$ emission events are generated to discern to what degree fires impact these anomalies. The purpose of this stratification, particularly stratification of days into anomalously high and low fire events, is to isolate the effects of fires on clouds and/or weather. The remainder of this section will detail how data stratification is accomplished. First, a variable is chosen for analysis (such as $CF$). Next, this variable as well as the variable(s) that are used to stratify the variable are filtered to include only the region of interest. As the Aqua satellite does not record data for each gridcell at every time step, wherever a coordinate (latitude,longitude,time) is missing a value for a specific variable, the variable(s) it is being stratified by also has the value at that coordinate replaced

by a missing value (and vice-versa). Next, to focus on potential feedbacks fires may have on land, a land-sea mask is applied. Then, the daily regional anomaly for each variable is taken. Then, the 2003-2022 wildfire seasons are spliced together, which results in a roughly 3060-day time series. From this 3060 day time series, any days with no data are removed. Next, the average of each day of the wildfire season is removed from each data point in the distribution to give a time series of anomalies for each variable. Booleans that filter out days above or below a certain percentile for the stratification variables are then applied simultaneously. For each dataset, an empirical CDF is then calculated. Then, the means are differentiated from each other to determine if the stratification variable (such as $DM$) leads to a significant change in the variable anomaly in question. This process can be applied both for a regional average, or on a gridcell-by-gridcell basis. When this process is performed on a gridcell-by-gridcell basis, the Pearson cross correlation coefficient $r$ is determined by spatially correlating the stratified variables with one another. This helps determine if one change in a variable as a result of fires (or other factors) feedbacks onto another to cause a change in anomaly. **Figure 1** serves as a verification of the stratification method, as well as validation of GFED4 emissions data. Monthly cross correlation analysis (Supplement Section 1, **Figure S2**) as well as previous works (Wilmot et al., 2022; Schlosser et al., 2017; Cho et al., 2022) indicate that during large fire events, $AOD$ and/or particulate matter concentration are significantly larger compared to no fire conditions. The significant increase in $AOD$ over most of the southwestern US supports the assertion that GFED4 fire emissions are an acceptable indicator of large fire occurrences.

It should be noted that while this method allows for the comparison of meteorology during very similar weather conditions, it does not completely remove the possibility of random meteorological fluctuations within the stratification that can affect the anomalies. Therefore, if anomalies are found, causality is difficult to discern.

### 3.3    Regions of Interest

First, the region within the southwestern US in which the most significant fire emissions originate was discerned. Based on what is generally considered to be the time of year in which most wildfires occur in the western US (Urbanski, 2013; Urbanski et al., 2011), data was collected from June 1st-October 31st for the 2003-2022 time period. 2003-2022 was chosen as this is the time period in which Aqua satellite data is available for the fire season. Analysis was limited to fire seasons as opposed to the entire year so that the threshold for what constitutes a 90th percentile fire is increased. First, for each gridcell, the 2003-2022 seasonal average daily $DM$ emissions was taken. The portion of the southwestern US that had the largest 2003-2022 seasonal average daily $DM$ emissions is the region that shall be referred to as "northern California" (nCA), which is highlighted in the blue box in **Figure 1a**. The reason for limiting $DM$ data to this region is again to ensure that the threshold for 90th percentile $DM$ is kept high. The nCA region is characterized by temperate forests along the coastline, in the far north, as well as the east. Agricultural lands are scattered throughout about every gridcell in nCA, with higher concentrations in the central valley as well as the coastal north. Grasslands are also found throughout most grid cells in this region, with higher concentrations in central CA. The dominant contributor of $DM$ in this region is the temperate forests in the north (**Figure S3**). At this time of year, predominant wind patterns in nCA would favor transportation of smoke from these fires to northern Nevada. During the fire season, northwesterlies tend to blow across nCA towards northern Nevada, and south westerlies blow through the central

valley and Sierra Nevada range (Zaremba & Carroll, 1999; LeNoir et al., 1999). Therefore, the expectation is for the majority of wildfire aerosols to be concentrated in nCA, and neighboring northern/central Nevada. In differentiating $AOD$ anomalies on high nCA $DM$ days and $AOD$ anomalies on low nCA $DM$ days, $AOD$ is found to be anomalously positive in both nCA and Nevada (**Figure 1b**), confirming this hypothesis. However, there are also significant $AOD$ anomalies throughout the entire region. For reasons that will be explained in **Section 4.1**, the main analysis will still be relegated to northern CA and Nevada. From this point forward, the focus will be on the effects of the fires in the blue box in **Figure 1a** (nCA) on the area highlighted in the green box (nCA-NV) in **Figure 1b**.

## 3.4 Heating Rate

Aerosol shortwave heating rate of the atmosphere $SWH_{aer}$ was calculated using

$$\frac{\partial T}{\partial t} = SWH_{aer} = \frac{g}{c_p} \cdot \frac{\Delta F_{aer}}{\Delta p} \tag{3}$$

where $t$ is time in days, $g$ is gravity, $c_p$ is the heat capacity at constant pressure, $F_{aer}$ is the shortwave radiative effect of the aerosols, and $p$ is pressure. $F_{aer}$ itself was derived from the CERES SYN1deg-Day downward and upward shortwave radiative fluxes. $F_{aer}$ between two atmospheric layers is given by

$$F_{aer} = SWd_1 - SWu_1 - (SWd_2 - SWu_2) \tag{4}$$

where $SWd_1$ denotes downward shortwave flux at the higher layer, $SWu_1$ denotes upward shortwave flux at the higher layer, $SWd_2$ denotes downward shortwave flux at the lower layer, and $SWu_2$ denotes upward shortwave flux at the lower layer.

## 4 Results

### 4.1 High & Low Surface Relative Humidity Stratification

The fingerprints of a traditionally-defined semi-direct effect where aerosols coincide with clouds would entail an anomalous warming of the cloud layer, and a corresponding decrease in $RH$. However, the meteorological conditions around which fires tend to occur need to be considered. As previously stated, large fires tend to occur during fire weather, which includes hot, dry, and windy conditions (Varga et al., 2022). Hot and dry conditions themselves are associated with high pressure anomalies in this region (**Figure S4**). Therefore, these fire weather conditions need to be "filtered out" as much as possible to isolate any potential semi-direct effects. Therefore, in addition to $DM$, variables need to be stratified by a second variable to account for the influence of meteorology on $P$, $CF$, and cloud properties. Fire season data was stratified by high (75th percentile) vs low (25th percentile) $T_s$, $RH_s$, $U_s$, and surface pressure to determine which variable was associated with the largest $DM$, and successfully filtered out fire weather condition anomalies. The 75th/25th percentiles were chosen for the potential second stratification variables as opposed to extremes (90th/10th percentiles) to ensure a robust number of data points, and to have a dataset that is more representative of common conditions in the region. **Figure 2** depicts CDFs for meteorological conditions

and $DM$ under high $RH_s$ extremes ($RH_s75$) and low $RH_s$ extremes ($RH_s25$) in the entire southwestern US. $RH_s$ was chosen as the second stratification variable, as stratifying nCA $DM$ by high ($RH_s75$) and low $RH_s$ conditions ($RH_s25$) and differentiating the means of these distributions yields a significant $DM$ anomaly of $\Delta DM = -1.04\text{e-}4 \pm 3.5\text{e-}5$ kg m$^{-2}$ day$^{-1}$. The absolute value of this anomaly is an order of magnitude higher than the differences in mean $DM$ between high and low conditions of the other potential stratification variables (surface pressure, $T_s$, and $U_s$ ) (**Figure S4, Figure S5, and Figure S6**). This indicates that fire occurrence/fire emission are more dependent on $RH_s$ than these other fire weather variables. Low $RH_s$ extremes in the southwestern US are associated with significantly higher $T$ throughout the troposphere/surface, significantly reduced $RH$ throughout the troposphere/surface, and significantly lower $CF$, while high $RH_s$ extremes are associated with the opposite (**Figure 2**). This demonstrates a need to separate the effects of fires from the meteorological effects of low $RH_s$ extremes, as positive $DM$ anomalies are significantly more likely to occur on ($RH_s25$) days as opposed to ($RH_s75$) days, as which is expected as moisture, and moist plants, suppress the ability of fires to grow and be maintained (Minnich & Chou, 1997; Ford & Johnson, 2006). The immediate direct effect of BB aerosols tends to be a net cooling of the surface (Sakaeda et al., 2011; Abel et al., 2005). However, certain semi-direct effects, such as the burning off of low clouds, may overpower this effect, leading to a net surface warming. As the meteorological conditions associated with low $RH_s$ days are also hallmarks of a semi-direct effect (**Figure 2**), from here onward data will be stratified into four categories: one with high $DM$ and high $RH_s$ ($DM90,RH_s75$), one with low $DM$ and high $RH_s$ ($DM10,RH_s75$), one with high $DM$ and low $RH_s$ ($DM90,RH_s25$), and one with one with low $DM$ and low $RH_s$ ($DM10,RH_s25$). In differentiating the average of the variables on ($DM90,RH_s75$) days and ($DM10,RH_s75$) days, the effects of the meteorological conditions that come with high $DM$ extremes can be minimized. However, a caveat to this analysis is that it is possible that there may be a bias towards lower values of $RH_s$ in the $DM90$ datasets compared to $DM10$ datasets, as fire weather conditions can invigorate fire activity. Therefore, while this analysis removes a lot of weather variability as per **Figure 2**, it does not remove all of it and caution should be taken when interpreting the results. **Figure 3** demonstrates that during large fires, $AOD$ anomalies under both high and low $RH_s$ stratifications are significantly positive in the nCA-NV region. The increase in mean $AOD$ is larger under low $RH_s$ at $0.24 \pm 0.04$. The corresponding change under high $RH_s$ is $0.13 \pm 0.05$. As the $AOD$ is consistently significant only in the nCA-NV region under both stratifications, this region will be the focus of the study.

## 4.2 Vertical Distribution of Black Carbon and Absorption in nCA-NV Region

Freshly emitted $BC$ is highly hydrophobic, and as it ages it becomes less resistant to accumulating water droplets (Lohmann et al., 2020). $BC$ has an average lifetime of 1 week (Lohmann et al., 2020), and the aging process begins after 1-2 days (He et al., 2016). Furthermore, in a region with such low fire season wet deposition such as the southwest US, the $BC$ on average can live much longer than one week (Ogren & Charlson, 1983). Therefore, hydrophobic and hydrophilic $BC$ profiles are important to differentiate because they can give an idea of how long the $BC$ stays in the atmosphere, and it hints at how much $BC$ can contribute to indirect and semi-direct effects. **Figure 4** displays high compared to low $DM$ mass mixing ratio anomalies for $BC_{phi}$, $BC_{pho}$, and combined $BC$ on high and low $RH_s$ days. Significant positive anomalies of $BC$ mass mixing ratio are present from 950Pa-300 hPa for all types of $BC$ under both ($DM90, RH_s75$) and ($DM90, RH_s25$) con-

ditions compared to the corresponding low fire conditions. The most significant increase in $BC$ is from about 950-600 hPa for the $(DM90, RH_s75)$ days, and from 950-550 hPa for the $(DM90, RH_s25)$ days. Comparing the MERRA-2 $BC$ profiles to the CALIPSO $DM90$-$DM10$ months 2006-2021 smoke aerosol daytime and nighttime extinction coefficient profile, MERRA-2 places more absorbing aerosol below 700 hPa, while CALIPSO generally places more absorbing aerosol above 700 hPa (**Figure S7**). Therefore, it is important to note that CALIPSO profiles do not agree with MERRA-2 when it comes to the positioning of the smoke in the troposphere. However, as the MERRA-2 and CALIPSO profiles are not temporally consistent, the comparison between these profiles is not 1-1. Additionally, as the CALIPSO profiles are not temporally consistent with the rest of the data in this paper, their use is not preferred over the MERRA-2 profiles.

There is roughly an equal amount of $BC_{phi}$ and $BC_{pho}$ during both high and low $RH_s$ days, indicating that on these days there is roughly as much fresh and aged aerosol in the troposphere. This is important as the quantity of $BC_{pho}$ indicates that microphysical effects are possible as it suggests a large amount of CCN are present in the troposphere. Additionally, the presence of aged $BC$ indicates that the $BC$ can affect the atmosphere radiatively over the course of multiple days. To estimate the impact of these aerosols on the troposphere over time, a $SWH_{aer}$ profile was created from CERES radiative flux data (**Figure 5**). Shortwave profiles used to generate these heating rate profiles, along with LW profiles, can be found in **Figure S8**. Under both $(DM90, RH_s75)$ (**Figure 5a**) and $(DM90, RH_s25)$ (**Figure 5b**) $RH_s$ conditions compared to the corresponding low $DM$ conditions, there is a positive $SWH_{aer}$ anomaly from 850 hPa to the next highest pressure level in the CERES dataset, 500 hPa. For high $RH_s$, this corresponds to a heating rate of $SWH_{aer} = 0.041 \pm 0.016$ K day$^{-1}$, and for low $RH_s$ this corresponds to a heating rate of $SWH_{aer} = 0.093 \pm 0.019$ K day$^{-1}$. Spatially, the 850 hPa-500 hPa heating rate is significant over almost all grid cells in the region of interest where there is data, with the most positive heating rates over eastern nCA and eastern Nevada (**Figure 5c,d**).

It should be noted that aerosol absorption can be affected by water vapor in the atmosphere, which can cause swelling and lensing effects that increase absorption (Wu et al., 2018; Peng et al., 2016). Therefore, this possibility will be investigated in Section 4.3.

## 4.3 Responses in Temperature, Humidity, & Cloud Profiles

**Figure 6** displays 2003-2022 June-October nCA-NV vertical profiles of high minus low fire $T$ (**Figure 6a,e**) and $RH$ (**Figure 6c,g**) profiles. **Figure 6a-d** are stratified by high $RH_s$, while **Figure 6e-h** are stratified by low $RH_s$. In both **Figure 6a** and **Figure 6e**, the temperature anomalies in the 850 hPa to 300 hPa pressure level range are consistently significant and positive at around 1 K. Comparing **Figure 6** to **Figure 4**, the positive differences in temperature anomaly are generally consistent with the positive $BC$ anomalies. Also, the changes in $T$ from 850 hPa-500 hPa are spatially consistent with the 850-500hPa heating rate anomalies where data is available (**Figure 5c, Figure 5d**). Under both high $RH_s$ (**Figure 6c**) and low $RH_s$ (**Figure 6g**) conditions, $RH$ anomalies throughout the entire profiles are negative but are only consistently significant during high $RH_s$ extremes. The AIRS $CF$ profile under high $RH_s$ conditions (**Figure 6d**) demonstrates significant negative anomalies from 300 hPa-600 hPa that are consistent with significant negative $RH$ anomalies and significant positive $T$ anomalies. However, there

is an increase in $CF$ at 850 hPa (**Figure 6d**). This pressure level corresponds to the highest concentration of $BC_{phi}$ (**Figure 4c**), and perhaps this indicates at this pressure level there is cloud seeding occurring. For the low $RH_s$ profile, there is only a significant negative cloud anomaly close to the surface at 925 hPa (**Figure 6h**).

Aside from temperature, the other potential factor that could affect $RH$ is that of specific humidity, which is analogous to water mass mixing ratio $M_{H_2O}$. **Figures 6b,f** depicts the effect of fires on $M_{H_2O}$ anomalies under high $RH_s$ and low $RH_s$ conditions respectively. There is no significant anomaly under high or low $RH_s$ conditions but is consistently positive at 700 hPa and below. Furthermore, the changes in the $RH$ profile follow the changes in the $T$ profile as opposed to the $M_{H_2O}$ profile, implying the positive $T$ anomalies generally dominate the change in $RH$ anomalies. The insignificant change in $M_{H_2O}$ also casts doubt that water vapor is affecting the absorption of the aerosols in any significant way.

While these profiles provide a general overview of how $T$, $M_{H_2O}$, $RH$, and $CF$ are changing over the region of interest, it is important to determine if these changes are consistent spatially with one another, as well as whether the changes coincide with $BC$ anomalies. As the $T$, $RH$, and $CF$ anomalies are strongest during high $RH_s$ days, the focus from here will be on the meteorological effects of high $DM$ on high $RH_s$ days. **Figures 7-11** depict the effect of fires on the spatial distributions of $BC$, $T$, $M_{H_2O}$, $RH$, and $CF$ anomalies at each AIRS pressure level up to 200 hPa on under high $RH_s$ conditions. The positive MERRA-2 $BC$ anomalies in **Figure 7** correlate positively and significantly with MODIS $AOD$ for each pressure level between 925 hPa-300 hPa (**Figures 7b-h**), and are spatially consistent with positive AIRS $T$ anomalies (**Figure 8**). Shifting attention to **Figure 9**, there appear to be significant negative anomalies in $M_{H_2O}$ in northeastern Nevada from 700 hPa-400 hPa, and significant positive anomalies over grid cells associated with large fires (**Figure 1a**) in the lower troposphere (925 hPa-850 hPa). Comparing these changes in $T$ and $M_{H_2O}$ spatially to changes in $RH$ (**Figure 10**), it appears that changes in $T$ tend to dominate changes in $RH$ over CA, western NV, and southern NV while changes in $M_{H_2O}$ appear to contribute to the negative $RH$ anomaly in northeastern NV. Additionally, the positive $M_{H_2O}$ at 850 hPa appears to mitigate the negative $RH$ anomalies at the same level, which may explain why $BC$ appears to be able to act as a CCN at this level but not others: $RH$ does not decrease enough to prevent clouds from forming. The increase in $M_{H_2O}$ has a myriad of possible explanations. It may be due to emission of moisture from the burned vegetation (Jacobson, 2014; Dickinson et al., 2021), from lofting of water vapor from the surface to higher levels of the atmosphere (Yu et al., 2024), or from moisture advection due to a change in wind vectors from the northeastern part of Nevada towards CA (**Figure S9**). This scattered significant increase in $M_{H_2O}$, being relegated to a few gridcells in few pressure levels, is not generally spatially consistent with the changes in $SWH_{aer}$ (**Figure 5c**), especially compared to the spatial distribution of $BC$ (**Figure 7**), further indicating that lensing effects are not the dominant contributor to the increase in aerosol absorption. Viewing **Figure 11c**, the increase in $CF$ at 850 hPa appears to be driven predominantly by a few significant and large coastal $CF$ anomalies. This indicates that there is an increase in shallow marine clouds at this pressure level, while clouds at other pressure levels are generally being suppressed. **Figure 11** demonstrates that significant negative $CF$ anomalies are generally spatially consistent with negative $RH$ anomalies from 700-400 hPa. The significant negative $CF$ anomalies in northeastern Nevada that correspond with significant negative $RH$ anomalies, but not significant

positive $T$ anomalies, at 700 hPa and higher indicate that the difference in clouds in this region is specific humidity dependent. This may be due to a transport of moisture outside of these grid cells due to anomalously positive southeastern wind speed anomalies in some of these grid cells (**Figure S9b**) that advect moisture towards southern California and southern Nevada, however further scrutiny is warranted to confirm this. It is not known if these wind speed anomalies are related to $T$ anomalies to the west, or if these wind speed anomalies in this region are an artifact. Changes in wind vectors are further analyzed in

supplement section 3. As the change in $T$ is the more robust signal over all parts of the troposphere, the changes in $T$ will be the focus of the remainder of the paper.

## 4.4 Changes in Cloud Type, Precipitation, and Shortwave Flux

With AIRS data indicating that large fires are associated with enhanced $T$, as well as lower $RH$ and $CF$, it is essential to determine how liquid vs ice clouds are impacted, and what the corresponding impacts on $P$ and radiative balance are. **Figure**

**12** displays CDFs for nCA-NV regional average variable anomalies during high $DM$/low $RH_s$ days (solid red), low $DM$/high $RH_s$ days (dashed red), high $DM$/low $RH_s$ days (solid blue), and low $DM$/low $RH_s$ days (dashed blue). **Figure 12a** and **Figure 12b** demonstrate that during high $RH_s$ extreme days, the effect of fires on the liquid water cloud fraction $CF_{lw}$ distribution and cirrus cloud fraction $CF_{cir}$ distribution is a significant shift towards a preference for negative anomalies. The effect of the large fires creates an average -0.04 $\pm$ 0.02 $CF_{lw}$ anomaly, and an average -0.05 $\pm$ 0.04 $CF_{cir}$ anomaly under

high $RH_s$ conditions. In addition, MODIS total $CF$ shifts by -0.07 $\pm$ 0.05 under the same stratifications. Precipitation also shifts significantly by -0.3 $\pm$ 0.23 mm day$^{-1}$. However, these shifts are significant only for high $RH_s$ extreme days (**Figure 12**). The explanation why the distribution shifts farther towards negative anomalies when anomalously large fires occur during high $RH_s$ compared to low $RH_s$ extremes lies in **Figure 2**. During low $RH_s$ days, $RH$ throughout the troposphere is already significantly lower than normal conditions (**Figure 2e**), as temperatures throughout the troposphere are already high (**Figure**

**2c**) and atmospheric water vapor content is low. This creates conditions of negative $CF$ anomalies (**Figure 2f**). Therefore, further increasing the already high $T$ should not lead to significantly lower cloud fraction as $RH$ is already low, and clouds require 100% $RH$ to form. This can also be explained by the $RH$ profile in **Figure 6g**, which demonstrates through most parts of the troposphere that $RH$ is not significantly lowered during fires. However, during low $DM$/high $RH_s$ days, **Figure 12** demonstrates that conditions are favorable for clouds and rain. This is because during these high $RH_s$ extremes, $T$ is lower and

$RH$ is high. Therefore, when anomalously large fires introduce a positive $T$ anomaly, the drop in $RH$ is significant enough to reduce the chances of seeing positive cloud/precipitation anomalies. In response to the higher probability of negative cloud fraction anomaly, the probability that SW radiation will be reflected into space decreases. This reduction in top of atmosphere shortwave flux leads to a net increase in cloud only (all-sky minus clear-sky) top of atmosphere radiative forcing $TOA_{cld}$ (**Figure 12f**). Though it should be noted that this increase is not significant, it is significant and positive over much of the

region marked by a decrease in $CF$ (**Figure 13e,h**), with a significant spatial cross correlation of $r = -0.67$. Regional all-sky SW and LW responses can be found in **Figure S10**.

**Figure 13** displays composite differences between meteorological variables on high $DM$/high $RH_s$ and low $DM$/high$RH_s$ days for each gridcell over the entire southwestern US. **Figures 13a,b** display the composite differences in cloud layer (850 hPa$\geq p \geq$300 hPa) temperature $T_{CL}$ and cloud layer relative humidity $RH_{CL}$. These plots depict that $T_{CL}$ significantly increases almost everywhere across California and Nevada, with the most significant increase in the green box (the nCA-NV region). The differences in $T_{CL}$ correlate significantly with differences in $AOD$ at $r = 0.72$ across the entire southwestern US. The decreases in $RH_{CL}$ have a very similar spatial distribution to $T_{CL}$, with the strongest decreases in the nCA-NV region. Again, this correlates significantly with $AOD$ with $r = -0.55$ over the entire southwest. The differences in all these variables across the southwestern US correlate significantly with $AOD$, supporting the assertion that aerosols concurrent with fires are associated with warming and drying. Of note are the changes in $T_s$ and $P$, which are two variables intrinsically related to fire duration. Spatially correlating $P$ with $RH_{CL}$ yields a significant, but notably weaker, correlation of $r = 0.44$, implying a relationship between the negative $P$ anomalies and the biomass burning aerosols. However, it should be noted that though the regional $P$ anomaly is significant and negative, that it appears to be dominated by just strong changes in just a few gridcells. $T_s$ correlates significantly with $AOD$ over the southwestern US, with $r = 0.51$, and is generally spatially concurrent with increases in $T_{cl}$ with $r = 0.72$. The equivalent for **Figure 13** for low $RH_s$ days is given in **Figure S11**. Of note for this supplementary figure is that there are weak, but significant and widespread, negative $CF$, $RH$, and $P$ anomalies over nCA and eastern Nevada, despite not being significant in the regional average (**Figure 12c,e**). This implies that the meteorological anomalies seen during high $RH_s$ days are also prevalent on low $RH_s$ days, but weaker and less widespread due to the lower availability of moisture.

While cross correlations indicate that there is a statistically significant relationship between fires and meteorology, practical significance needs to be established as well. The effect sizes of high $DM$ emissions on nCA-NV regional averages of the variables in **Figure 12** and **Figure 13** are depicted in **Figure 14**. For high $RH_s$ extremes (**Figure 14a**), the anomalously large fires are associated with a moderate-to-strong effect size on most of the relevant variables. **Figure 14b** demonstrates that during low $RH_s$ conditions, anomalously large fires are associated weak-to-no effect size on the relevant variables, aside from $T_{cl}$ in which fires have a very strong effect size on. It should be noted that effect size does not imply causality, but instead only quantifies how different the mean of a distribution is when a single variable is changed.

## 4.5 Cloud Microphysical Effects

Up to this point, we have investigated how cloud fraction and type differ during large fires. Aerosols from wildfires may also influence clouds via microphysical effects, which are investigated in this section. High fire emissions under high $RH_s$ conditions are associated with non-significant differences in microphysical variables (**Figure 15**). Spatial maps of high minus low fire $R_{eff}$ and $LWP$ under high $RH_s$ conditions show a mix of areas with positive and negative changes, most of which are not significant (**Figure S12**). Although there is a small tendency for negative $R_{eff}$ anomalies to occur in Nevada and a small tendency for negative $LWP$ anomalies to occur in nCA and western NV. Since negative $R_{eff}$ anomalies can affect precipitation,

the spatial distribution of $R_{eff}$ anomalies (**Figure S12, Figure S13**) was compared to the spatial distribution of $P$ anomalies (**Figure 13, Figure S11**) under high compared to low $DM$ conditions. Significant negative $R_{eff}$ anomalies were not found to be spatially consistent with significant negative $P$ anomalies under either high or low $RH_s$ conditions. This casts doubt on wildfires in this region creating microphysical suppression of $P$.

There are significant regional changes in liquid $R_{eff}$ and $LWP$ under low $RH_s$ conditions (**Figure 15, Figure S13**). Liquid $R_{eff}$ significantly increases under these conditions, which is contrary to what one would expect as a response to increased $AOD$ (Twohy et al., 2021; Conrick et al., 2021; Fan et al., 2016). One possible explanation for this increase in $R_{eff}$ is that $R_{eff}$ is directly proportional to temperature (Martins et al., 2011), and perhaps the effects of the $T$ anomaly dominate over the condensation of new droplets onto $BC_{phi}$. Alternatively, this increase may be driven by changes in atmospheric dynamics, as increased updraft strength and enhanced turbulence could lead to increased coalescence (Khain, 2009). Coincident with the strongest increase in $R_{eff}$ (at the northernmost coast of California) under these conditions is a significant negative (upward) pressure velocity anomaly from 1000 hPa-925 hPa, which implies that an increase in upward convection near the surface may be a factor of the increase in $R_{eff}$, as an upward pressure velocity should increase droplet lifetime (**Figure S14**). It is also noted that there are negative pressure velocity anomalies under high $RH_s$ conditions from 1000 hPa-850 hPa (**Figure S15**), and this corresponds with an increase in $R_{eff}$ near the Bay Area.

Comparing high to low fire conditions, $LWP$ under simultaneously low $RH_S$ conditions shows a significant decrease (**Figure 15c**). This significant negative $LWP$ anomaly may be due to the negative $RH_{cl}$ anomaly (**Figure S11b**), as lower saturation of the air would reduce liquid water within clouds. This decrease in $LWP$ may be of importance, as $LWP$ scales positively with cloud albedo (Han et al., 1998). Therefore, this decrease in $LWP$ may contribute to an increase in absorbed solar radiation at the surface. In summary, while the nCA fires significantly inject aerosols into the troposphere, these aerosols do not appear to generally act as CCN, and instead contribute to a positive $T$ anomaly that burns off clouds. This may be because $BC$ is generally more hydrophobic compared to other aerosols, and instead the radiative effects of the aerosol dominate.

## 5   Discussion

The results of this paper indicate that large fires in nCA are concurrent with significant amounts of absorbing aerosols, which themselves are associated with a shortwave heating rate of 0.04-0.09 Kday$^{-1}$. This heating rate contributes to positive $T$ anomalies in the region that are concurrent with large fires, however the extent of this contribution is unknown. When the fires occur during high $RH_s$ conditions, the positive $T$ anomalies (**Figure 8, Figure 13a**) are associated with significant negative $RH$ anomalies in the low, mid, and high cloud layers (850 hPa-300 hPa) at the 90% confidence interval (**Figure 10, Figure b**). These negative $RH$ anomalies are associated with a reduction of clouds, which is associated with significant negative $P$ anomalies in the nCA-NV region. These negative $CF$ anomalies are also associated with an increase in $TOA$ radiative flux (**Figure 13h**), despite a decrease in $CTH$ (**Figure 13f**). In short, wildfires in nCA are associated with region wide negative

$CF$ anomalies that are cause by positive $T$ anomalies. Aerosols emitted from biomass burning contribute to these positive $T$ anomalies through shortwave absorption, indicating that the traditionally defined aerosol-cloud semi-direct effect is a possible explanation for the decrease in clouds. Furthermore, the $T$, $RH$, $CF$, anomalies correlate significantly with positive $BC$ and $AOD$ anomalies (**Figure 7, Figure 8, Figure 10, Figure 11, Figure 13**), further supporting the assertion that aerosols are contributing to these anomalies. However, it is unknown to what extent the aerosols contribute to the $T$ anomalies observed,

and therefore to the negative $RH$, $CF$, and $P$ anomalies. One possible source of noise is wind. **Figure S9** depicts a positive wind speed in northern Nevada that may be influencing cloud cover over that part of the region, and it is unknown if this signal has anything to do with the positive $T$ anomalies. Additionally, wildfires are associated with an increase in sensible heat flux from the combustion of biofuels, which may contribute to the positive $T$ anomaly as well (Dickinson et al., 2021). Furthermore, random weather variations within the stratification may also create anomalies favorable enhanced fire activity, which would

increase $DM$, making causality difficult to discern. However, another study that utilized a similar methodology to this paper to analyze the effects of large fires using combined aircraft observations and a climate model indicates the possibility that the aerosols in this study are a significant contributor to the negative $CF$ anomalies (Thornhill et al., 2018). They ran the Met Office Unified Model using aircraft observations of $AOD$ and BB aerosol properties. They compared meteorological variables in high vs low fire emission conditions over South America and found a clear sky shortwave heating rate of the low-to-mid

troposphere that is larger (0.2 K day$^{-1}$), but comparable, to the heating rates calculated in this paper. This was also associated with a higher $BC$ mass mixing ratio, and a significant negative $CF$ anomalies of around 0.08, which is a similar anomaly to the $0.07 \pm 0.05$ MODIS $CF$ anomaly observed in this study during high $RH_s$ conditions (**Figure 12c**). Though not related to fire, an aircraft observational study of anthropogenic $BC$ over the Bay of Bengal found a $BC$ heating rate of around 0.5 Kday$^{-1}$ (Kant et al., 2023), which further demonstrates that $BC$ can be associated with atmospheric warming. Furthermore, the results

of this study are consistent with numerous other satellite observational studies over the tropics and subtropics that demonstrate that aerosols associated with wildfires are shortwave absorbing and can contribute to burn-off of clouds, resulting in a positive radiative forcing (Wilcox, 2012; Kaufman et al., 2005; Ackerman et al., 2000; Hansen et al., 1997). Additionally, the reduction in $CF$ and $P$ is consistent with the results of Chen et al. (2014), which was a biomass burning modelling experiment conducted over the United States. However, their proposed mechanism for these decreases was a change in convection due to the

distribution of warming of the aerosols. Concerning the increase in $M_{H_2O}$ above sites of fire emission in **Figure 9b,c**, this is consistent with a recent study that found comparable results (Yu et al., 2024), however they found more water vapor higher in the troposphere than this paper. Additionally, it is noted that the observed microphysical effects of the BB aerosols in this paper, namely the lack of a regional decrease in $R_{eff}$, contrast to another observational study that overlaps with the region of interest in this paper (Twohy et al., 2021). An important note about that study, however, is that it only sampled the 2018

wildfire season while this study focuses on the entire 2003-2022 time span.

The results of this paper highlight that it is necessary to understand the contribution of biomass burning aerosols to the anomalies that favor enhanced fire weather. If the aerosols are a significant contributor to these anomalies, this can create a positive feedback loop where large fires emit copius amounts of $BC$, warm the atmosphere, reduce cloud cover, suppress $P$, and there-

fore intensify fire activity. As this potential feedback would prolong wildfires, it would therefore also prolong poor air quality conditions inside the southwestern US (Liu & Peng, 2019; O'Neill et al., 2021; Schlosser et al., 2017), as well as other parts of the country (Hung et al., 2020). Significant reductions in nCA $P$ may prolong the wildfire season further into autumn (Goss et al., 2020), and increases in $T$ as well as decreases in $RH$ may create conditions more favorable for more fires to ignite and grow (Varga et al., 2022). Additionally, the negative $P$ anomalies and/or positive $T_s$ anomalies in this paper occur in heavily populated regions in the southwestern US, including: the San Francisco Bay Area, Humboldt County in California, and Washoe County in Nevada. Therefore, it is essential to further investigate the relationship between anomalously large fires in the region and the local meteorology, as if the fires are contributing to these meteorological anomalies, this would dictate an increased need for a curtailment of $CO_2$ emissions (Ma et al., 2021; Touma et al., 2021) and better land management practices (DellaSala et al., 2022; Minnich et al., 2000; Minnich, 2001), as climate change and land mismanagement have both contributed to the large fires in nCA in recent years. Additionally, the confirmation that these $BC$ anomalies are associated with a positive heating rate anomaly is enough to advocate for these changes, as the fires are worsening already warm western US weather. Furthermore, as large fires are projected to become more commonplace throughout the 21st century due to these factors (Flannigan et al., 2013; United Nations Environment Programme, 2022), the results of this paper will become more relevant over time as today's 90th percentile fire emission conditions become more common throughout the 21st century.

Overall, to determine if the fires are significantly contributing to the negative $RH$, $CF$, and $P$ anomalies, it is essential to run a climate modelling experiment where $BC$ is increased over the region of interest, and to quantify the effects of this increased $BC$ on these meteorological variables.

*Code availability.* Code used to process satellite data will be made available at the following GitHub repository: https://github.com/jgome222/Northern-California-large fires-Associated-with-Decrease-in-Cloud-Cover-Over-the-Southwestern-US

*Data availability.* All datasets utilized in this analysis are available online. MODIS datasets are available via the 787 NASA Level-1 and Atmosphere Archive & Distribution System (LAADS) Distributed Active Archive 788 Center (DAAC) at https://ladsweb.modaps.eosdis. nasa.gov/archive/allData/61/. CERES datasets can be found at https://ceres.larc.nasa.gov/. AIRS data is available via NASA's Earth Science Data 794 extremes (ESDS) program at https://www.earthdata.nasa.gov/. CALIPSO datasets are available at the Atmospheric Science Data Center (ASDC) at https://asdc.larc.nasa.gov/. GFED4 fire emission data is archived on the GFED4 web page at https://www.globalfiredata. org/. MERRA-2 data can be found on the Goddard Earth Sciences Data and Information Services Center (GES DISC) website at https://disc.gsfc.nasa.gov/datasets?project=MERRA-2.

## Appendix A

| Symbol | Definition | Dataset Derived From | Name of Product(s) Used |
|---|---|---|---|
| $BC$ | Black Carbon | MERRA-2 | $BC$PHILIC, $BC$PHOBIC |
| $DM$ | Fire dry matter emissions | GFED4 | DM, daily_fraction |
| $AOD$ | Aerosol Optical Depth | MODIS | Aerosol_Optical_Depth_Land_Ocean_Mean |
| $M_{H_2O}$ | Water Mass Mixing Ratio | AIRS | H2O_MMR_D |
| $T$ | Temperature | AIRS | Temperature_D |
| $T_s$ | Surface Temperature | AIRS | SurfAirTemp_D |
| $RH$ | Relative Humidity | AIRS | RelHum_D |
| $RH_s$ | Surface Relative Humidity | AIRS | RelHumSurf_D |
| $CF$ | Cloud Fraction | MODIS | Cloud_Fraction_Mean |
|  |  | AIRS | FineCloudFrc_D |
| $CF_{cir}$ | Cirrus Cloud Fraction | MODIS | Cirrus_Fraction_Infrared |
| $CF_{lw}$ | Liquid Water Cloud Fraction | MODIS | Cloud_Retrieval_Fraction_Liquid |
| $CTH$ | Cloud Top Height | MODIS | Cloud_Top_Height_Mean |
| $P$ | Precipitation | GPCP | precip |
| $SWH_{aer}$ | Aerosol Shortwave Heating Rate | CERES | adj_all_sw_dn, adj_all_sw_up, adj_naer_sw_dn, adj_naer_sw_up |
| $F_{aer}$ | Shortwave aerosol radiative forcing | CERES | Same as above variable |
| $TOA_{cld}$ | Cloud-only Net Top of Atmosphere Flux | CERES | adj_all_sw_dn, adj_all_sw_up, adj_all_lw_up adj_clr_sw_dn, adj_clr_sw_up, adj_clr_lw_up |
| $SWu$ | Shortwave aerosol upward flux | CERES | adj_all_sw_up, adj_clr_sw_up |
| $SWd$ | Shortwave aerosol downward flux | CERES | adj_all_sw_dn, adj_clr_sw_dn |
| $U_s$ | Surface Wind speed | CERES/GEOS | sfc_wind_speed |
| Liquid $R_{eff}$ | Liquid Cloud Effective Radius | MODIS | Cloud_Effective_Radius_Ice_Mean |
| Ice $R_{eff}$ | Ice Cloud Effective Radius | MODIS | Cloud_Effective_Radius_Liquid_Mean |
| LWP | Liquid Water Path | MODIS | Cloud_Water_Path_Liquid_Mean |
| IWP | Ice Water Path | MODIS | Cloud_Water_Path_Ice_Mean |

**Table A1.** Definition of variables that were derived from satellite observational datasets, as well as the instrument and dataset they are derived from.

| Symbol | Definition |
| --- | --- |
| nCA | Northern California |
| nCA-NV | Northern California-Nevada |
| US | United States |
| BB | Biomass Burning |
| OA | Organic Aerosol |
| CA | California |
| SW | Shortwave |
| AAOD | Absorbing Aerosol Optical Depth |
| LW | Longwave |
| TOA | Top of atmosphere |
| CCN | Cloud Condensation Nuclei |
| CDF | Cumulative Distribution Function |

**Table A2.** Definitions of abbreviations found throughout the paper that are not associated with a dataset.

| Descriptor | Definition |
|---|---|
| $(DM90)$ | Variable stratified by 90th percentile fire dry matter emission anomaly days in nCA |
| $(RH_s75)$ | Variable stratified by 75th percentile surface relative humidity anomaly days in nCA-NV |
| $(DM10)$ | Variable stratified by 10th percentile fire dry matter emission anomaly days in nCA |
| $(RH_s25)$ | Variable stratified by 25th percentile surface relative humidity anomaly days in nCA-NV |
| $(DM90, RH_s75)$ | Variable stratified by 90th percentile fire dry matter emission anomaly days in nCA and 75th percentile surface relative humidity anomaly days in nCA-NV |
| $(DM10, RH_s75)$ | Variable stratified by 10th percentile fire dry matter emission anomaly days in nCA and 75th percentile surface relative humidity anomaly days in nCA-NV |
| $(DM90, RH_s25)$ | Variable stratified by 90th percentile fire dry matter emission anomaly days in nCA and 25th percentile surface relative humidity anomaly days in nCA-NV |
| $(DM10, RH_s25)$ | Variable stratified by 10th percentile fire dry matter emission anomaly days in nCA and 25th percentile surface relative humidity anomaly days in nCA-NV |
| cl | Cloud layer (850-300 hPa) average of variable |
| s | Variable measured at the surface |
| pho | Hydrophobic aerosol |
| phi | Hydrophilic aerosol |
| aer | radiative forcing variable calculated from all-sky minus clear sky products (aerosol only) |
| cld | radiative forcing variable calculated from all-sky minus no aerosol products (cloud only) |
| $\Delta$ | Difference in variable under different fire and/or relative humidity conditions |

**Table A3.** Definitions of subscripts and other descriptors for variables.

*Author contributions.* J.L.G. conceived the project, designed the study, performed data analysis and wrote the paper. R.J.A. performed analyses, and wrote the paper. K.L. advised on methods.

*Competing interests.* The authors declare no competing interests.

*Acknowledgements.* R.J. Allen is supported by NSF grant AGS-2153486.

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

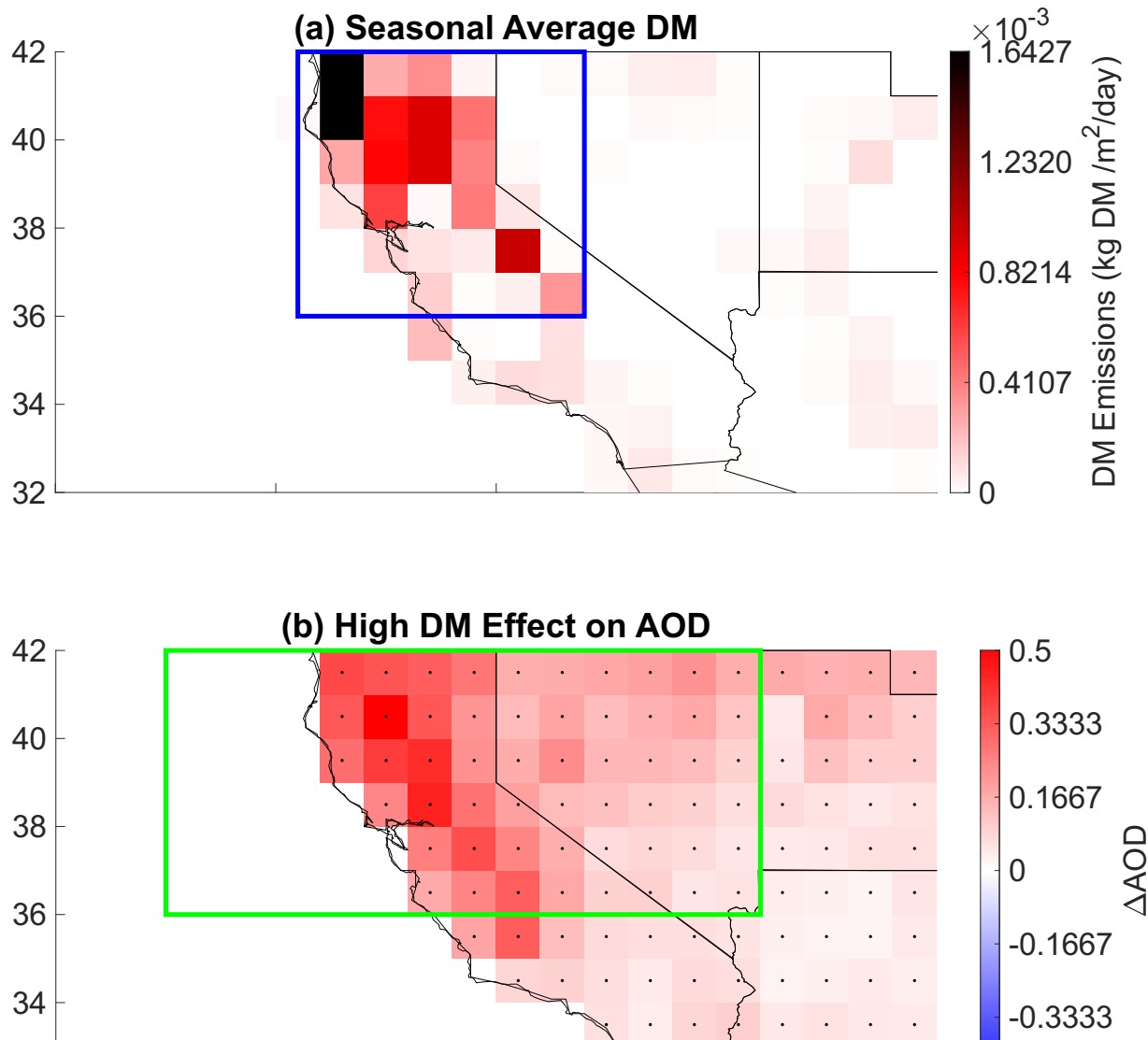

**Figure 1.** Distribution of fires and the corresponding aerosol optical depth $AOD$ anomaly impacts during the fire season. (a) 2003-2022 average daily fire dry matter $DM$ emissions for the southwestern United States during the fire season (June-October). Blue box signifies the nCA (northern California) region, where average daily fire emissions are the highest. (b) 2003-2022 June-October daily MODIS Aerosol optical depth (AOD) difference between average $AOD$ on 90th percentile $DM$ ($DM90$) and average $AOD$ on 10th percentile $DM$ ($DM10$) days within the 2003-2022 June-October time period. $\Delta AOD$ represents $AOD(DM90) - AOD(DM10)$. Green box symbolizes the nCA-NV (northern California-Nevada) region, where increases in $AOD$ and changes in cloud properties (**Figure 11**) are most significant. Black dots represent statistically significant differences at $90\%$ confidence according to a two-tailed test.

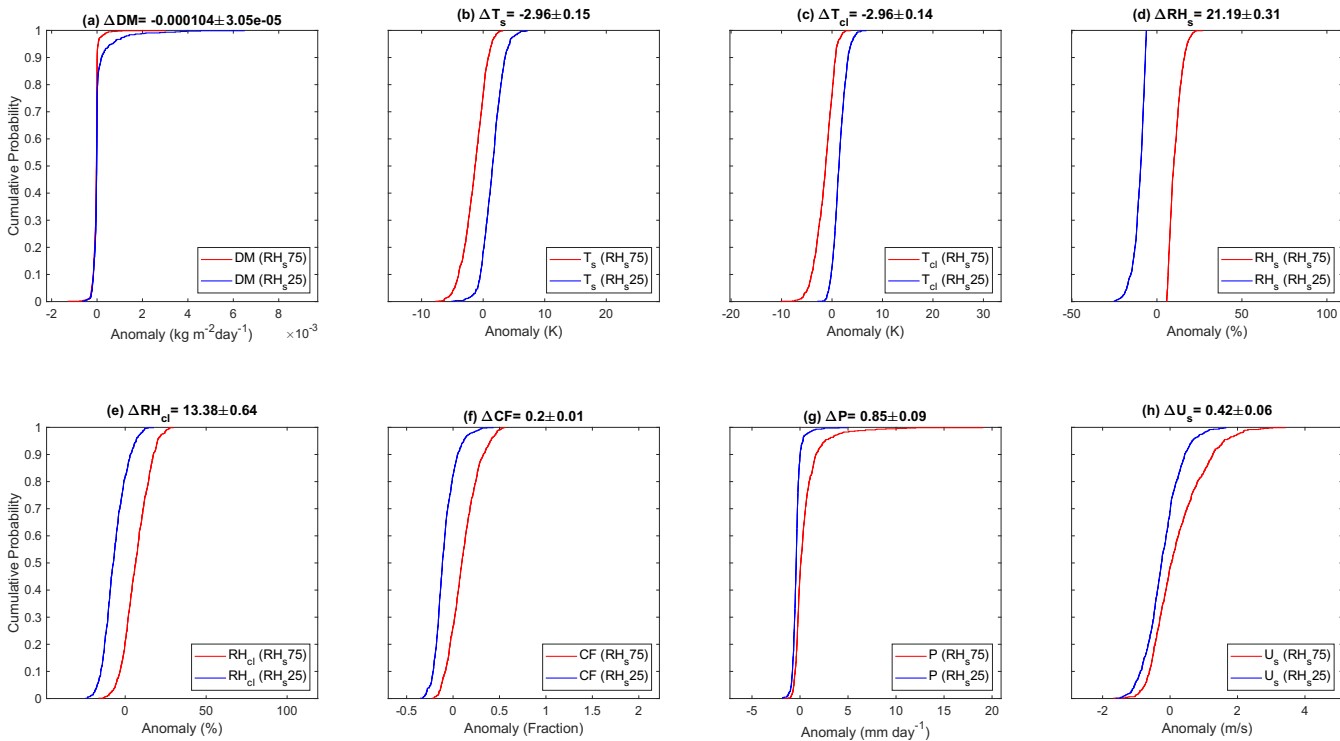

**Figure 2.** Dependence of meteorological variables on high versus low surface relative humidity $RH_s$ during the fire season. Regional average cumulative distribution functions (CDFs) for variable anomalies stratified by 75th percentile surface relative humidity ($RH_s75$) days (red) and 25th percentile ($RH_s25$) (blue) days within the 2003-2022 June-October time period. Variables depicted include (a) northern California (nCA) fire dry matter (DM) emissions, (b) southwestern US surface temperature $T_s$, (c) nCA-NV cloud layer (850-300 hPa) average temperature $T_{cl}$, (d) southwestern US surface relative humidity $RH_s$, (e) southwestern US cloud layer average relative humidity $RH_{cl}$, (f) southwestern US cloud fraction $CF$, (g) southwestern US precipitation $P$, and (h) southwestern US surface wind speed $U$. $\Delta$ represents the difference between the variable's average anomaly for $RH_s75$ and $RH_s25$ days.

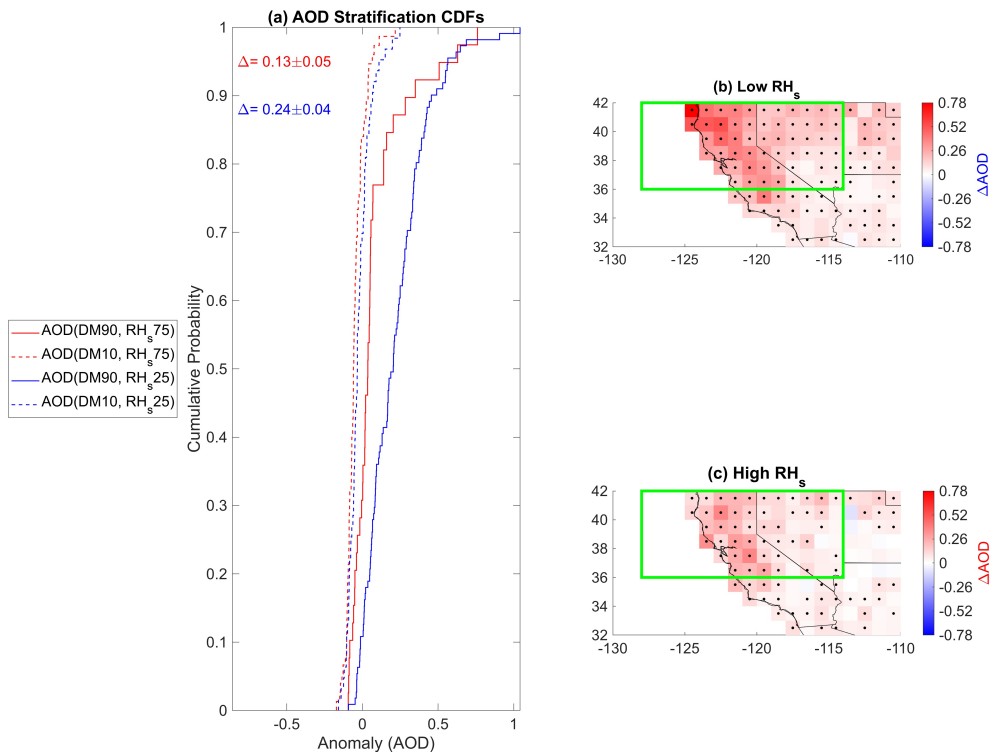

**Figure 3.** Difference in $AOD$ anomalies on high and low $RH_s$ days during the fire season. Daily nCA-NV $AOD$ anomalies stratified by nCA-NV $RH_s$ and nCA $DM$ extremes within the 2003-2022 June-October time period. (a) displays cumulative distribution functions for daily June-October 2003-2022 daily northern California-Nevada nCA-NV $AOD$ stratified by high (90th percentile) nCA $DM$ emissions and high nCA-NV $RH_s$ $AOD(DM90, RH_s75)$ (solid red line), low (10th percentile) $DM$ and high $RH_s$ $AOD(DM10, RH_s75)$ (dashed red), high $DM$/low $RH_s$ $AOD(DM90, RH_s25)$ (solid blue line), and low nCA $DM$/low $RH_s$ $AOD(DM10, RH_s25)$ (dashed blue line). The red $\Delta AOD$ represents the difference between the solid red and dashed red line $AOD(DM90, RH_s75)$-$AOD(DM10, RH_s75)$ and the blue $\Delta AOD$ represents the difference between the solid and dashed blue lines $AOD(DM90, RH_s25)$-$AOD(DM10, RH_s25)$. (b) Depicts a map of $AOD(DM90, RH_s25)$-$AOD(DM10, RH_s25)$. Pearson cross correlation coefficient $r$ between $\Delta AOD$ and nCA $DM$ emissions is depicted in the top left corner. (c) Depicts a map of average $AOD(DM90, RH_s75)$-$AOD(DM10, RH_s75)$. Black dots in (b),(c) represent statistically significant differences at the 90% confidence interval according to a two-tailed test.

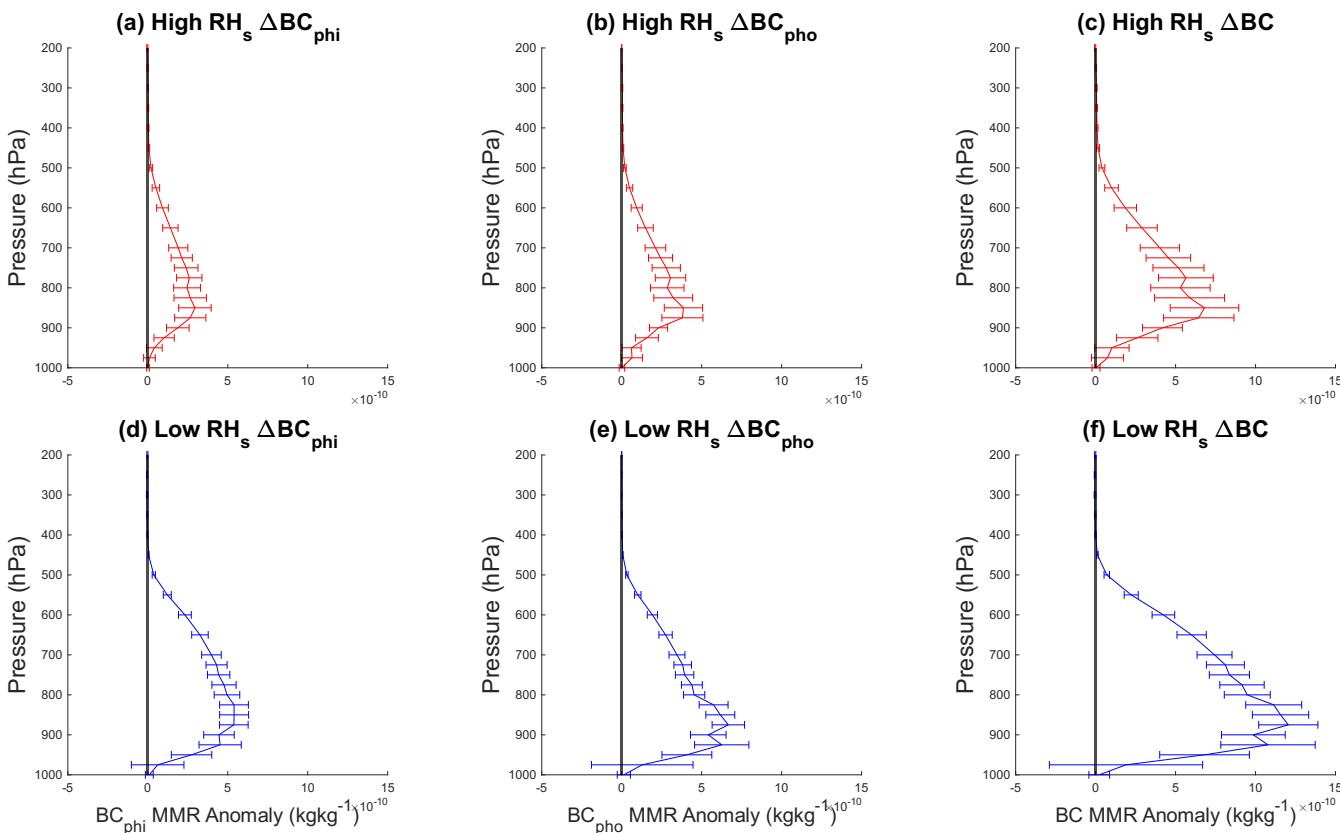

**Figure 4.** Difference in MERRA-2 black carbon $BC$ profiles on high vs low fire days stratified by differing $RH_s$ conditions in the nCA-NV region in the 2003-2022 June-October time period. Profiles of both aged hydrophilic black carbon $BC_{phi}$ (a,d) as well as freshly emitted hydrophobic black carbon $BC_{pho}$ (b,e) are depicted in addition to total $BC$ (c,f). All types of $BC$ have significant anomalies from 850-300 hPa under both high $RH_s$ (a-c) as well as low $RH_s$ conditions (d-f).

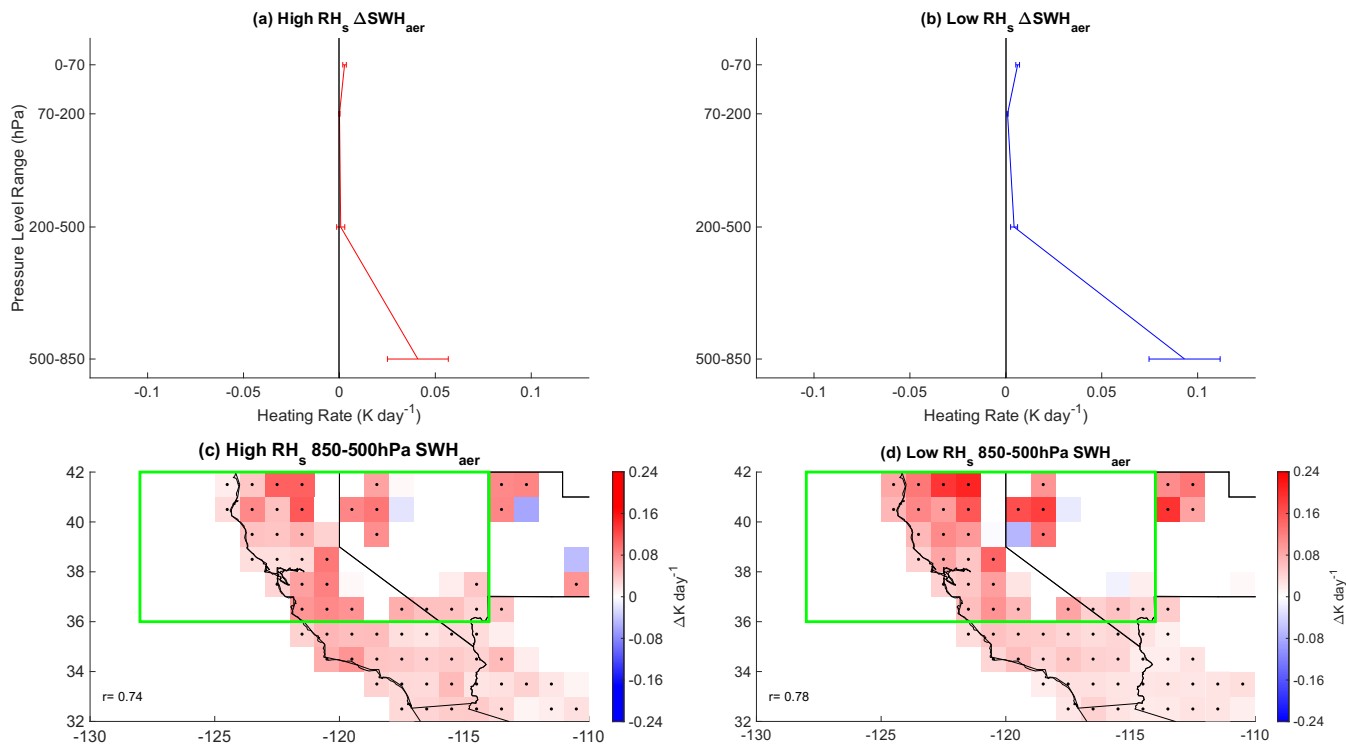

**Figure 5.** High minus low $DM$ days regional average aerosol-only shortwave heating rate $SWH_{aer}$ profiles under differing $RH_s$ conditions in the 2003-2022 June-October time period. There is a significant shortwave aerosol heating rate from 850-500 hPa under both high $RH_s$ conditions (a) as well as low $RH_s$ conditions (b). Also depicted are spatial maps for high minus low fire days (c) under simultaneously high $RH_s$ conditions and (d) under simultaneously low $RH_s$ conditions. Black dots represent statistical significance at the 90% confidence interval. $r$ represents the cross correlation between $SWH_{aer}$ and $AOD$.

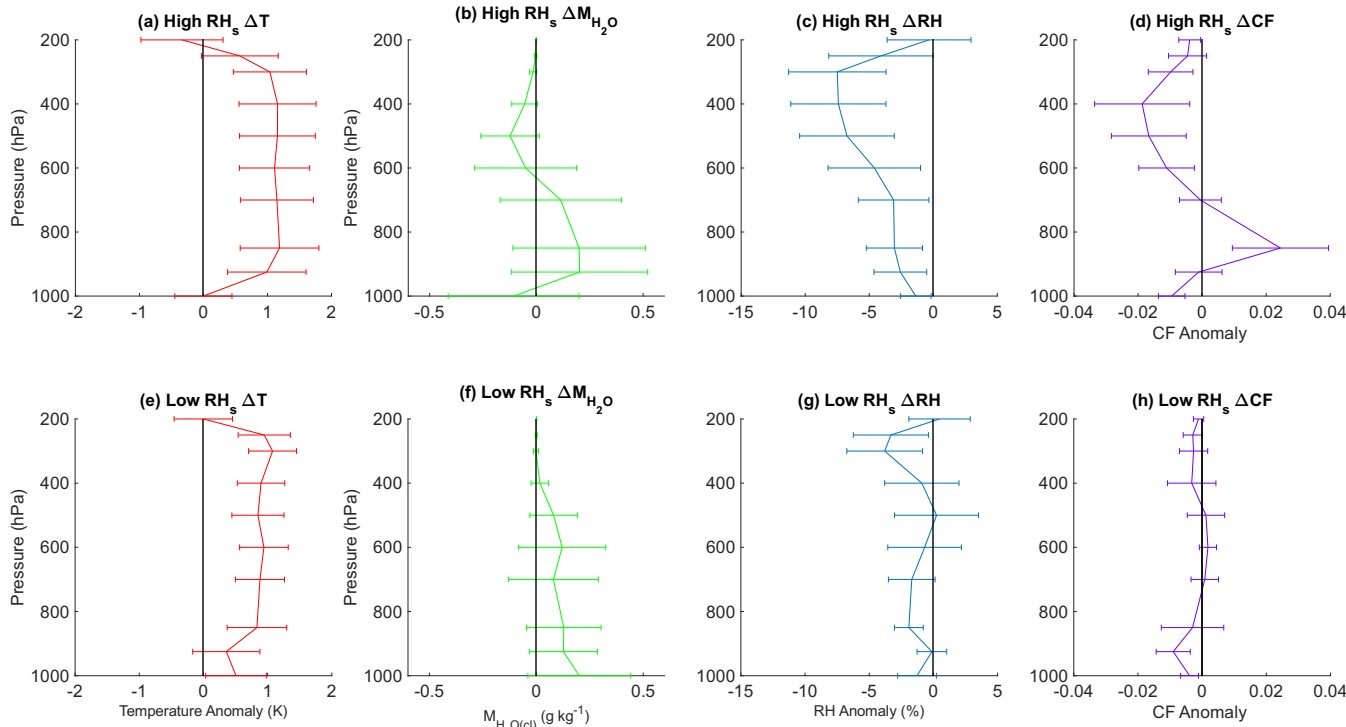

**Figure 6.** Responses in AIRS temperature $T$, water mass mixing ratio $M_{H_2O}$, relative humidity $RH$, and cloud fraction $CF$ profiles to large fires under high and low $RH_s$ extremes during the fire season. nCA-NV regional-temporal average differences in $T$, water mass mixing ratio $M_{H_2O}$ and relative humidity $RH$ profiles for under high minus low $DM$ conditions stratified by $RH_s 75$ (a-d) and $RH_s 25$ (e-h) in the 2002-2023 fire season (June-October) time period. Error bars represent the 90% confidence interval.

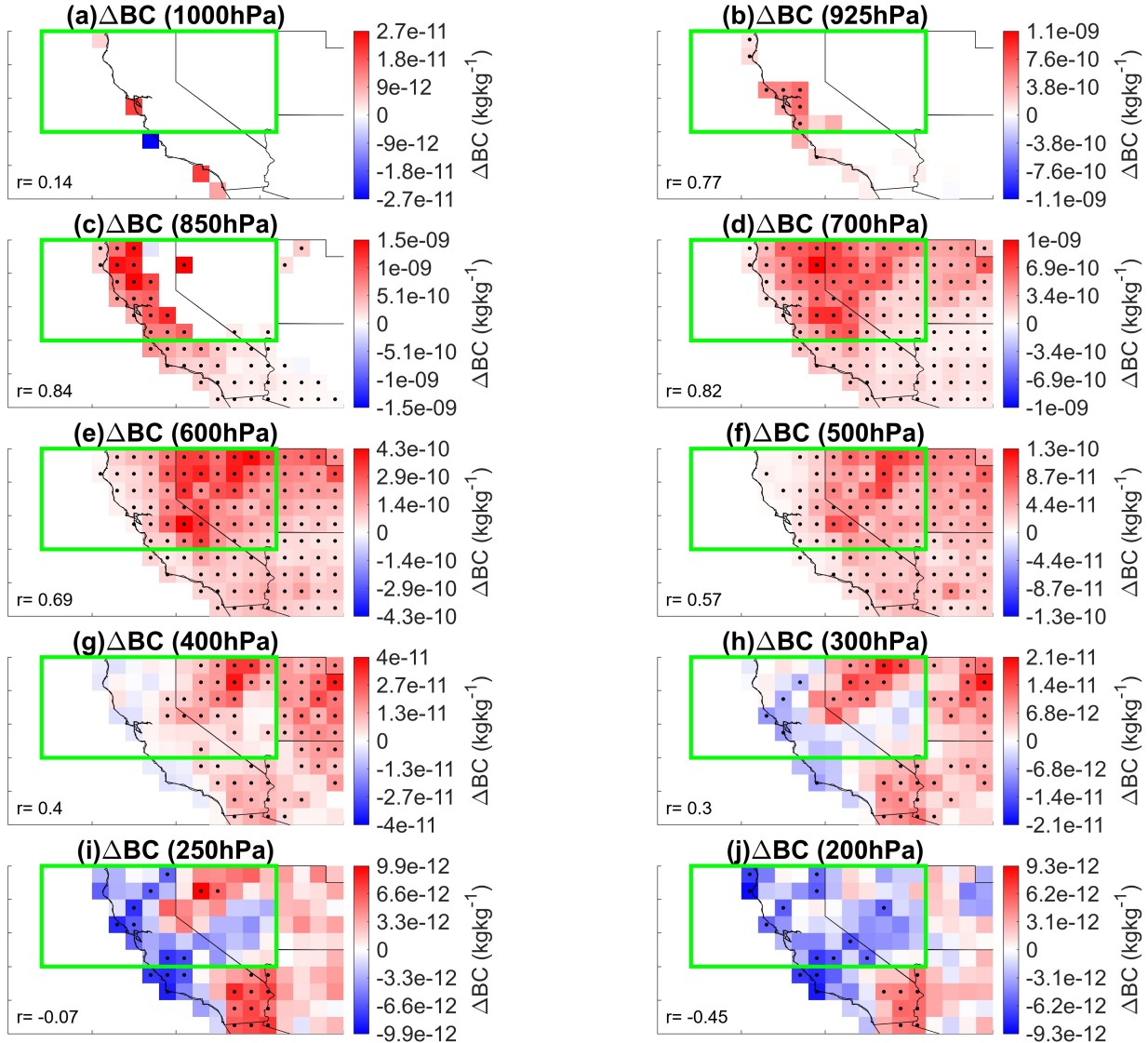

**Figure 7.** High minus low $DM$ days MERRA-2 $BC$ anomalies at all AIRS pressure levels from 1000 hPa to 200 hPa (a-j) under high $RH_s$ conditions in the 2003-2022 June-October time period. Black dots indicate statistical significance at the 90% confidence interval. $r$ values indicate spatial Pearson cross correlations between total $BC$ and MODIS $AOD$.

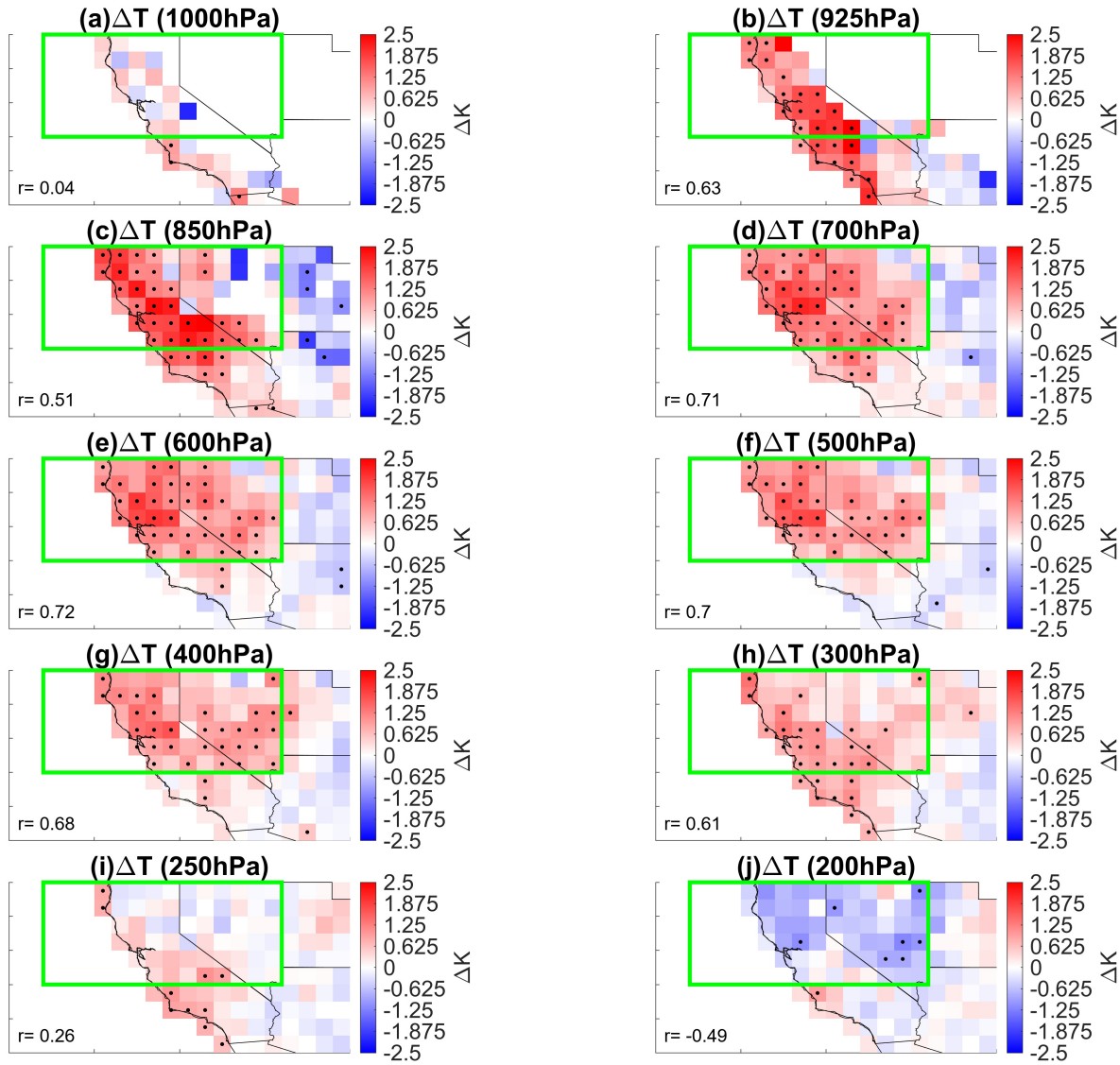

**Figure 8.** High minus low $DM$ days AIRS $T$ anomalies at all AIRS pressure levels from 1000 hPa to 200 hPa (a-j) under high $RH_s$ conditions in the 2003-2022 June-October time period. Black dots indicate statistical significance at the 90% confidence interval. $r$ values indicate spatial Pearson cross correlations between $T$ and MODIS $AOD$.

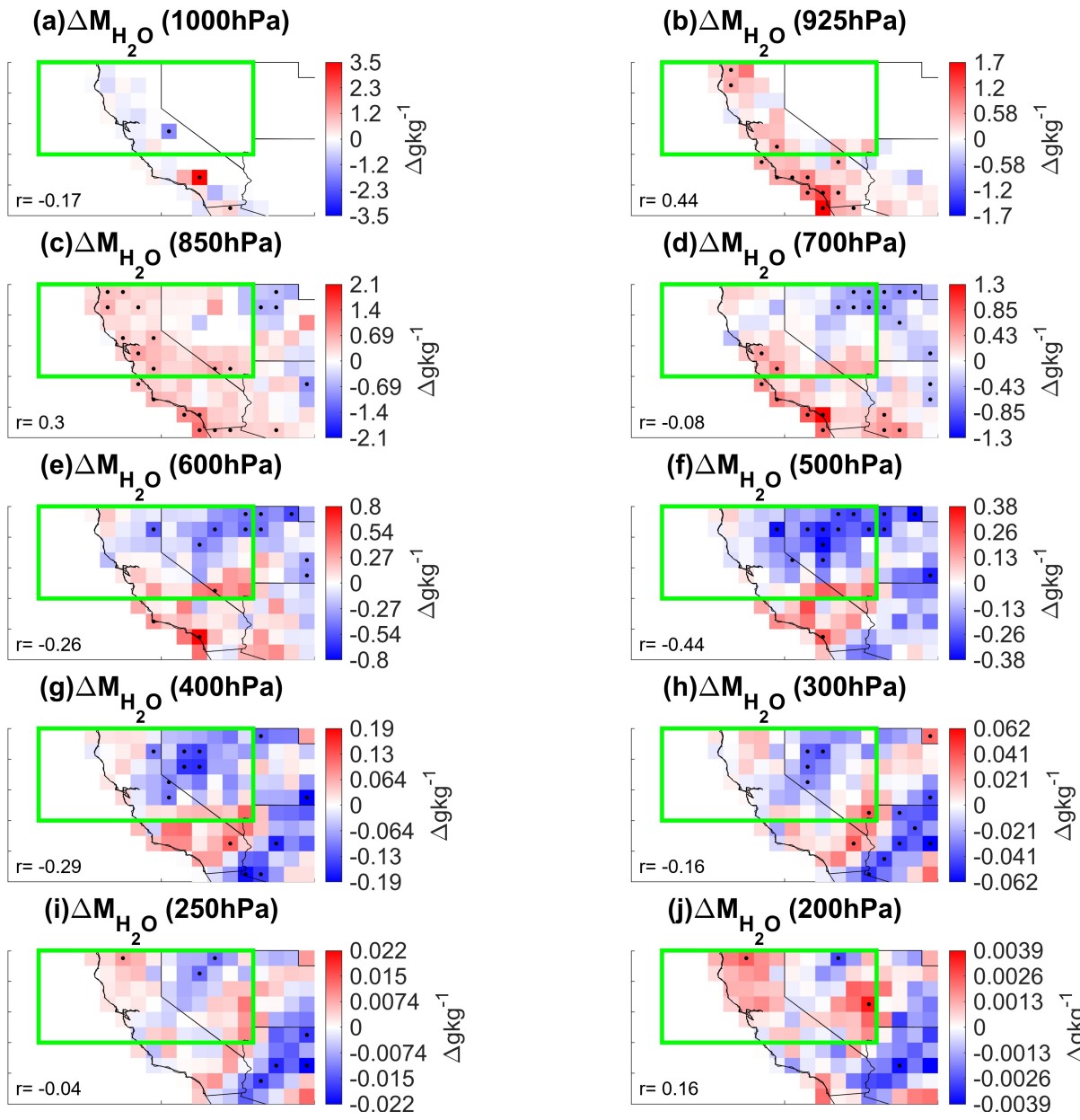

**Figure 9.** High minus low $DM$ days AIRS $M_{H_2O}$ anomalies at all AIRS pressure levels from 1000 hPa to 200 hPa (a-j) under high $RH_s$ conditions in the 2003-2022 June-October time period. Black dots indicate statistical significance at the 90% confidence interval. $r$ values indicate spatial Pearson cross correlations between total $M_{H_2O}$ and MODIS $AOD$.

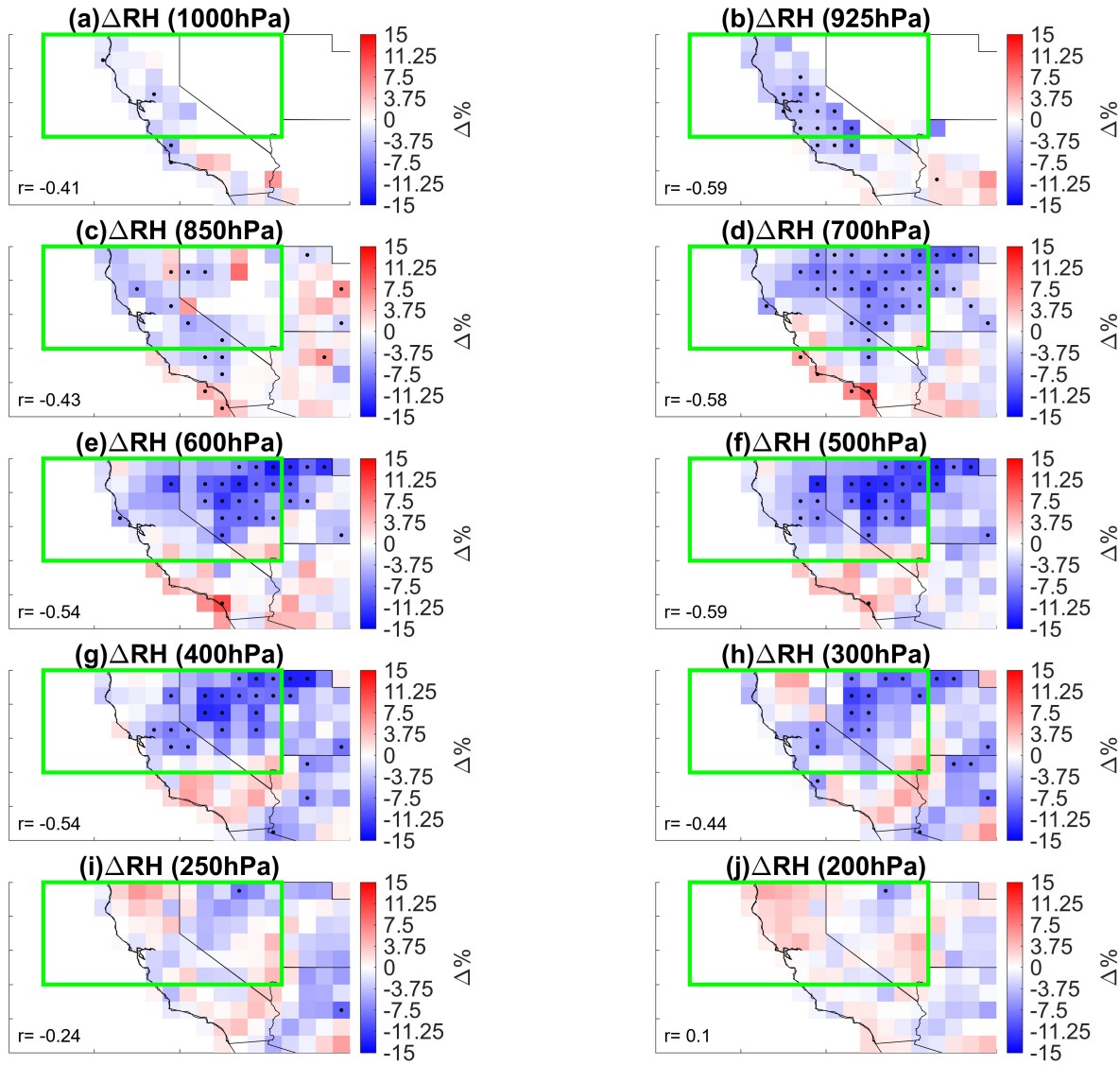

**Figure 10.** High minus low $DM$ days AIRS $RH$ anomalies at all AIRS pressure levels from 1000 hPa to 200 hPa (a-j) under high $RH_s$ conditions in the 2003-2022 June-October time period. Black dots indicate statistical significance at the 90% confidence interval. $r$ values indicate spatial Pearson cross correlations between $RH$ and MODIS $AOD$.

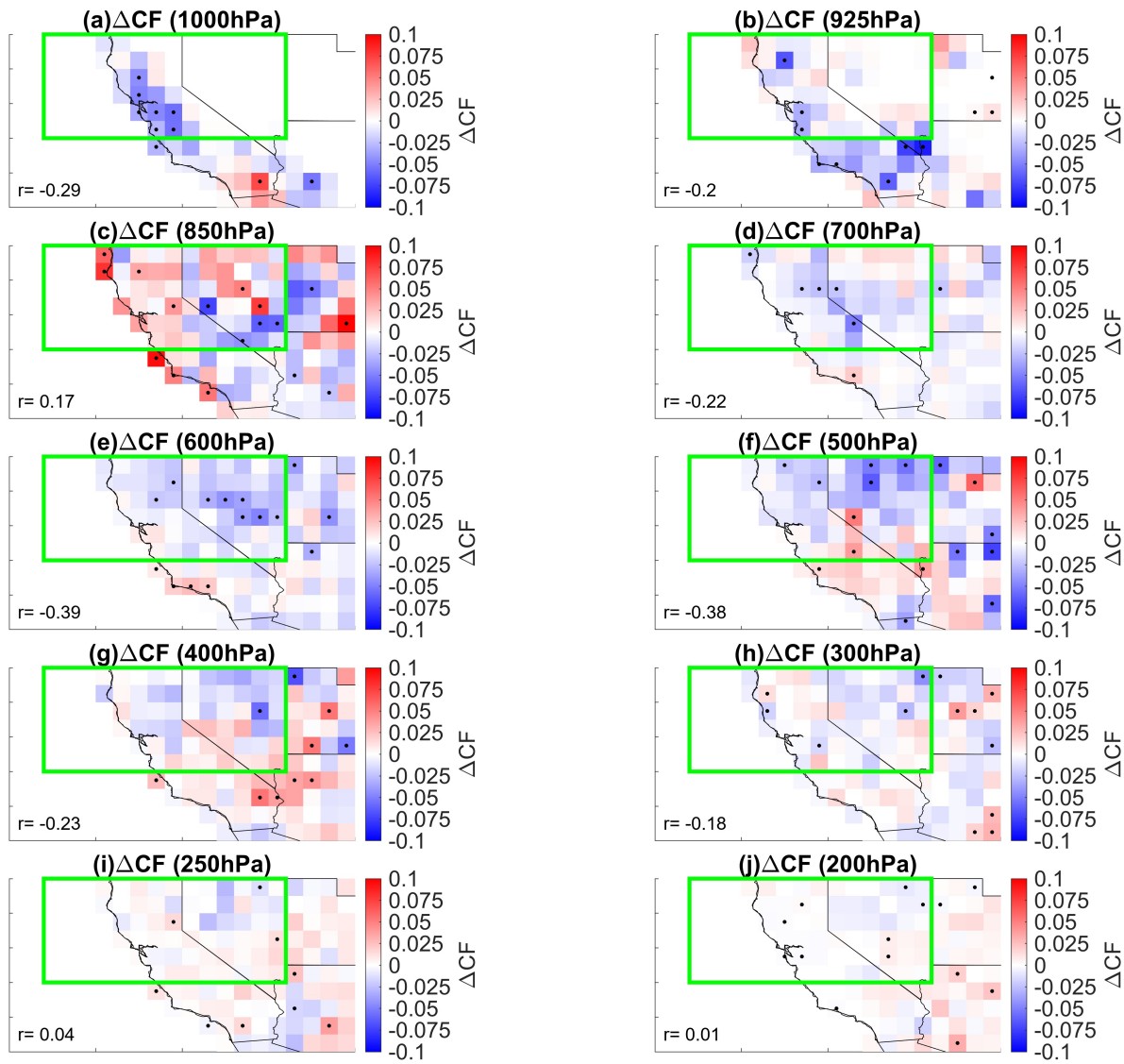

**Figure 11.** High minus low $DM$ days AIRS $CF$ anomalies at all AIRS pressure levels from 1000 hPa to 200 hPa (a-j) under high $RH_s$ conditions in the 2003-2022 June-October time period. Black dots indicate statistical significance at the 90% confidence interval. $r$ values indicate spatial Pearson cross correlations between $CF$ and MODIS $AOD$.

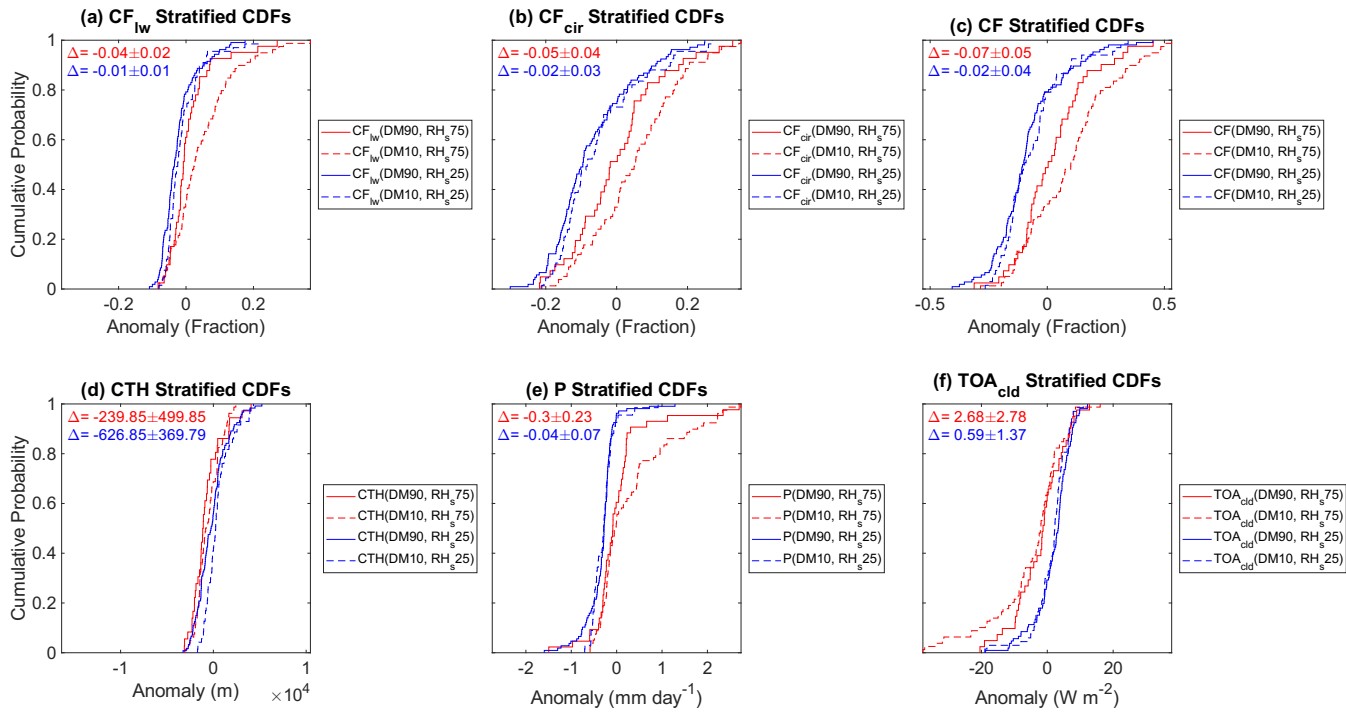

**Figure 12.** Dependence of meteorological variables on high versus low $RH_s$ and fires during the fire season. Empirical CDFs for regional average daily anomalies of meteorological variables over the nCA-NV region in the 2003-2022 June-October time period. Solid red line signifies variable anomalies are stratified by high nCA fire dry matter emission $DM$ and high nCA-NV $RH_s$ anomaly days ($DM90, RH_s75$). The dashed red line signifies variable anomalies are stratified by low $DM$ and high $RH_s$ anomaly days ($DM10, RH_s75$). The solid blue line represents variable anomalies are stratified by high $DM$ and low $RH_s$ anomaly days ($DM90, RH_s25$). The dashed blue line symbolizes variable anomalies are stratified by low $DM$ and $RH_s$ anomaly days ($DM10, RH_s25$). Variables depicted include (a) liquid water cloud fraction $CF_{lw}$, (b) cirrus cloud fraction $CF_{cir}$, (c) $CF$, (d)cloud top height $CTH$, (e) precipitation $P$, and cloud-only (all-sky minus clear-sky) net top of atmosphere flux $TOA_{cld}$. The red $\Delta$ represents the differences in the mean of the solid red and dashed red lines ($DM90, RH_s75$)-($DM10, RH_s75$). The blue $\Delta$ represents the differences in the mean of the solid blue and dashed blue lines ($DM90, RH_s25$)-($DM10, RH_s25$).

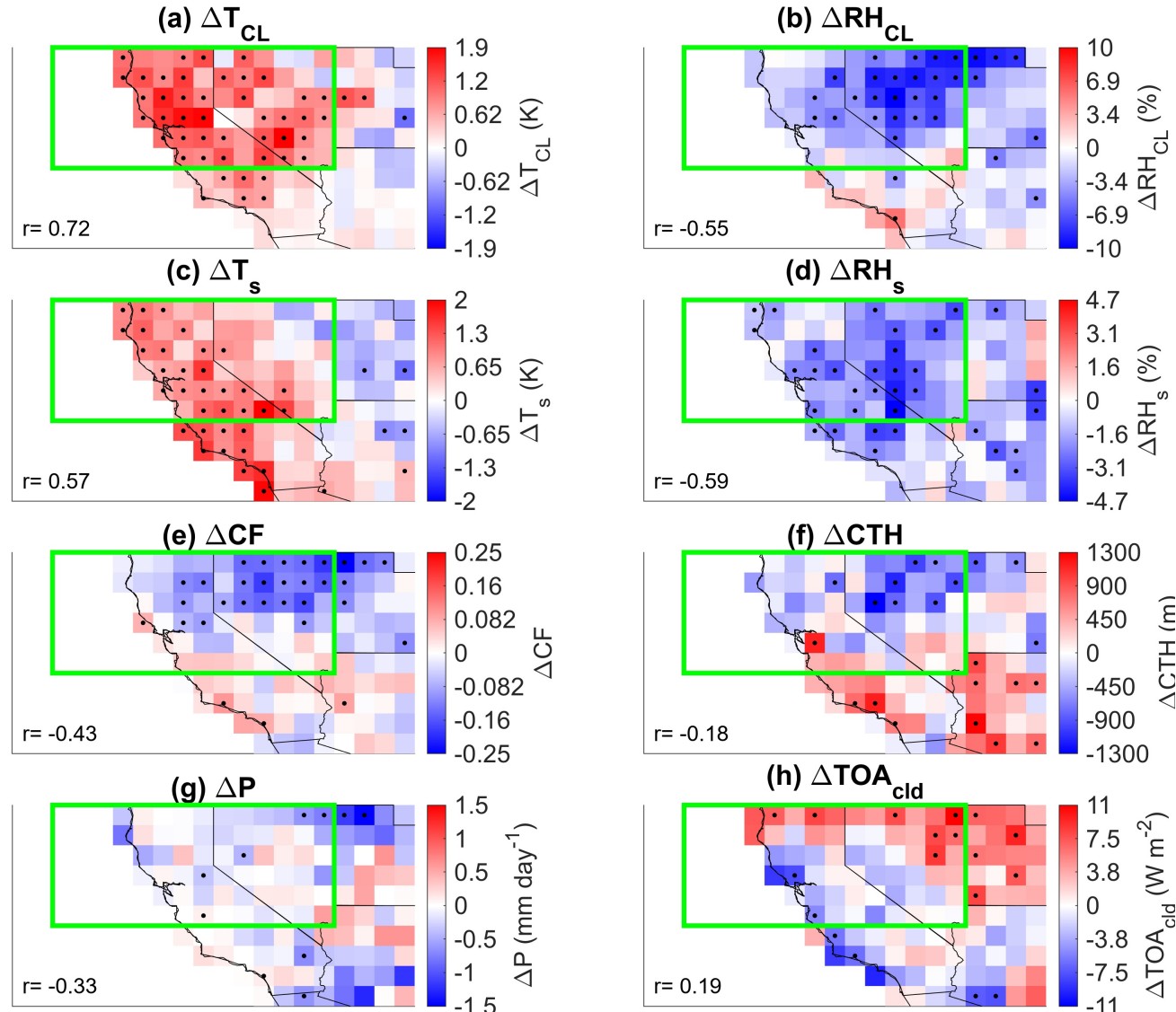

**Figure 13.** Meteorological responses under high versus low nCA $DM$ conditions with simultaneously high nCA-NV $RH_s$ during the fire season. Difference between average variable anomalies on high (90th percentile) nCA fire dry matter $DM$ emission days and low (10th percentile) nCA $DM$ emission days that occur on high nCA-NV $RH_s$ days in the 2003-2022 June-October time period. Variables include (a) 850 hPa-300 hPa average Temperature $T_{cl}$, 850 hPa-300 hPa average relative humidity $RH_{cl}$, (c) surface temperature $T_s$, (d) $RH_s$, (e) $CF$, (f) $CTH$, (g) $P$, and (e) $TOA_{cld}$. Black dots represent statistically significant differences at the 90% confidence interval according to a two tailed test. Pearson cross correlation $r$ values in each plot represent the spatial correlation between MODIS aerosol optical depth $AOD$ anomaly and the variable anomaly depicted in the figure. All values of $r$ are significant at the 90% confidence interval according to a two-tailed test.

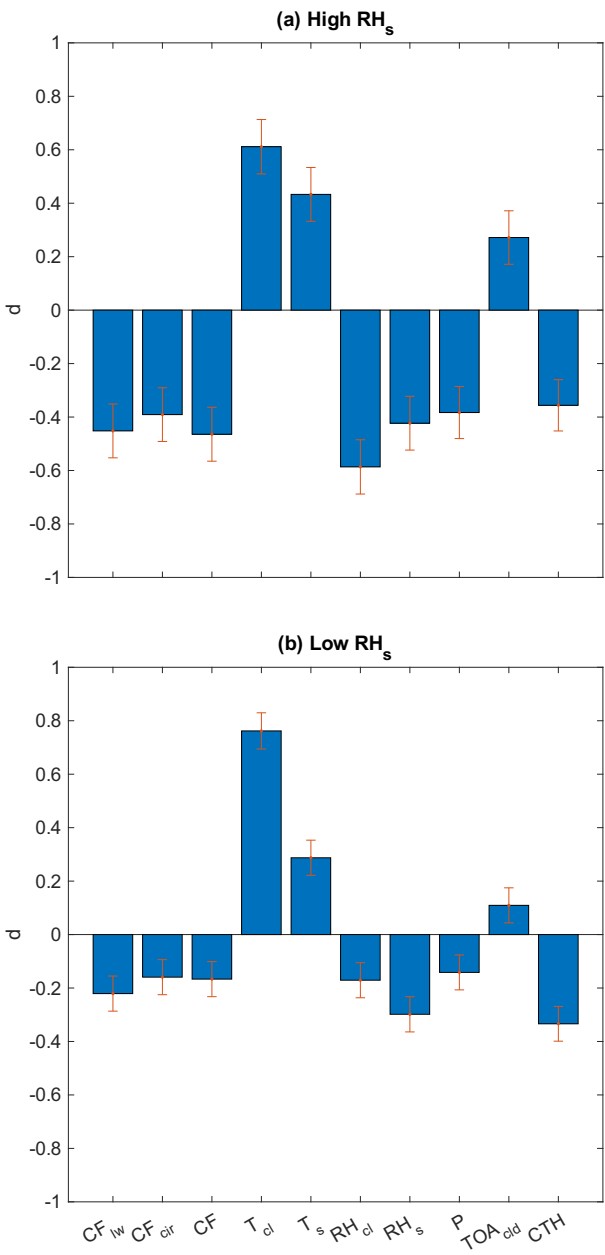

**Figure 14.** Effect size of large fires in nCA on the mean of various meteorological variables during the fire season. 2003-2022 June-October Cohen's d $d$ values for the difference between nCA-NV regional averages of variables on high $DM$ days minus low nCA $DM$ emission days that coincide with (a) high $RH_s$ and (b) low $RH_s$. For Cohen's d, values of 0.2 through 0.5 signify a weak effect size, values of 0.5 through 0.8 represent a moderate effect size, and values greater or equal to 0.8 signify a strong effect size. Red bars represent standard error.

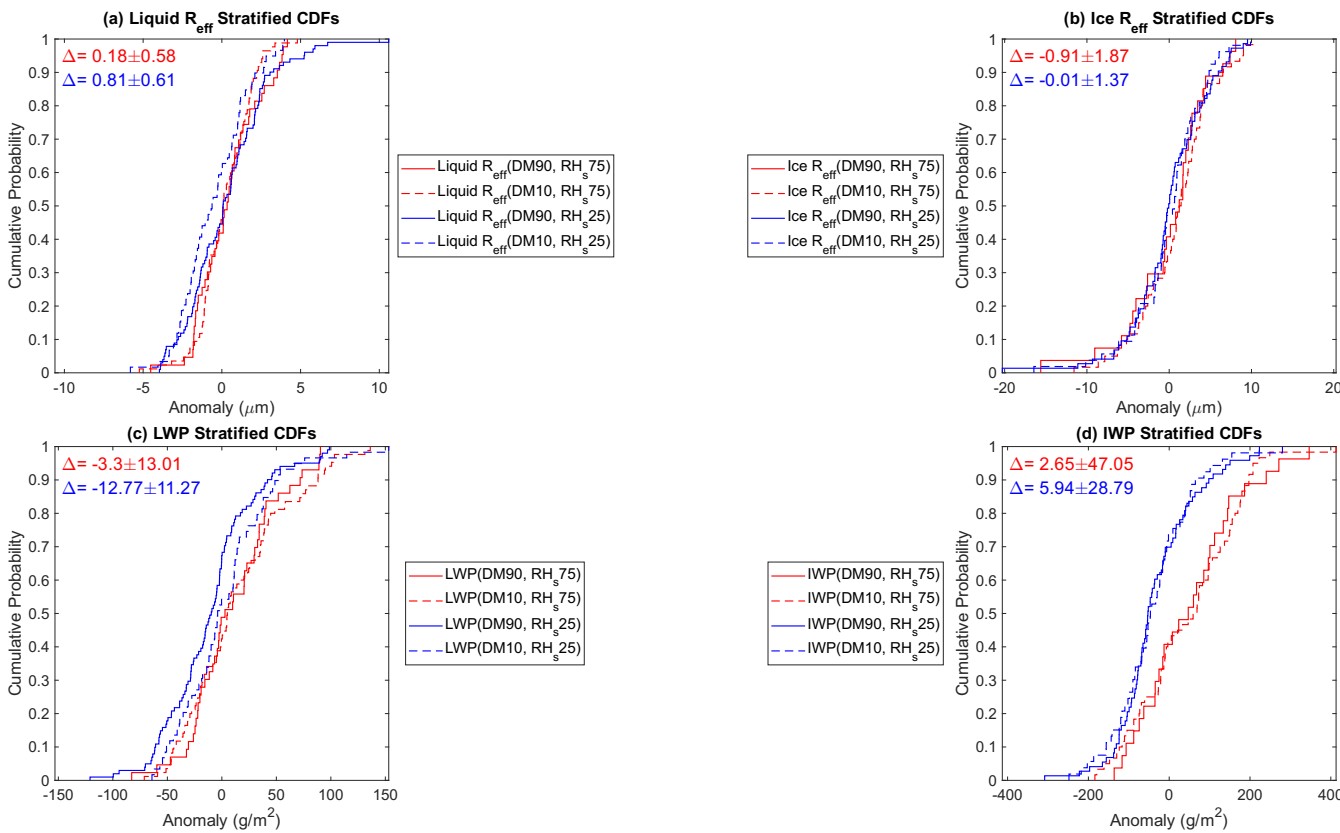

**Figure 15.** Dependence of microphysical variables to high versus low surface relative humidity $RH_s$ and fires during the fire season. Empirical CDFs for regional average daily anomalies of cloud microphysical variables over the nCA-NV region in the 2003-2022 June-October time period. Solid red line signifies variable anomalies are stratified by $(DM90, RH_s75)$. The dashed red line signifies variable anomalies are stratified by $(DM10, RH_s75)$. The solid blue line represents variable anomalies are stratified $(DM90, RH_s25)$. The dashed blue line symbolizes variable anomalies are stratified by $(DM10, RH_s25)$. Variables depicted include (a) liquid effective radius $R_{eff}$, (b) Ice $R_{eff}$, (c) liquid water path $LWP$, (d) and ice water path $IWP$. The red $\Delta$ represents the differences in the mean of the solid red and dashed red lines $(DM90, RH_s75)$-$(DM10, RH_s75)$. The blue $\Delta$ represents the differences in the mean of the solid blue and dashed blue lines $(DM90, RH_s25)$-$(DM10, RH_s25)$.