# Peer review of "California Wildfire Smoke Contributes to a Positive Atmospheric Temperature Anomaly over the Western United States"

_EGUsphere, 2023_

## Author Comment (AC1)

**Response to Reviewers**

We thank the reviewers for their helpful comments that have significantly improved this paper. We have taken all of these recommendations into consideration and have made major revisions to our manuscript. Below is the response to the reviewers' comments.

**Reviewer #1:**

The semi-direct effect. There needs to be a clear link between the presence of absorbing aerosol and changes to the cloud-field due to temperature. The vertical profiles of extinction show signs that smoke (and polluted dust) is enhanced in high fire-activity months. This dataset, as stated in the manuscript, is a monthly product on a very coarse grid. However, without coincident profiles of aerosol to directly compare against the increases in T, decrease in RH, and decrease in CF it is very difficult to make the SDE link robust.

Thank you for your comment. This criticism has led us to adopt a more robust approach to establishing a link between the biomass burning aerosols and the semi-direct effect.

We now utilize aerosol profiles that are coincident both spatially and temporally. The MERRA-2 M2I3NVAER reanalysis dataset provides aerosol mass mixing ratios of black carbon on every day utilized for the region of interest. Additionally, this dataset is high resolution (361 x 576), and can be interpolated to the 1 degree grid utilized in this paper.

While this dataset is a reanalysis dataset, it still utilizes satellite AOD observations and provides daily output of aerosol fields including black carbon associated with wildfires. Furthermore, this dataset is validated using observations (Buchard et al. 2015). As per the revision:

"Daily vertical black carbon aerosol mass mixing ratio profiles are derived from the M2I3NVAER data product (Global Modeling And Assimilation Office & Pawson, 2015; Buchard et al., 2015). This product estimates aerosol profiles by assimilating MODIS AOD into the GEOS5 model, which is radiatively coupled to the Goddard Chemistry, Aerosol, Radiation, and Transport (GOCART) aerosol module. The GOCART model includes biomass burning emissions from the NASA Quick Fire Emission Dataset (QFED) version 2.1, which provides daily biomass burning aerosol estimates (Buchard et al., 2015). These profiles were then validated using ground and satellite observations of aerosol profiles. This dataset has been previously used to determine effects of wildfire aerosols in other parts of the world (Raga et al., 2022; Nguyen et al., 2020)."

Additionally, we have added AIRS cloud fraction profiles. These profiles show a decrease in clouds coincident with increases in T as well as decreases in RH from 400hPa-600hPa. According to MERRA-2, there is also a significant increase in black carbon at these levels. We

now also include spatial maps at all AIRS levels to ensure that the changes in T, RH, and CF are consistent at all pressure levels (Figures 7-11).

I don't think this would be an issue if SDE was stated as a possible mechanism, but as it is the main conclusion (and title) of the manuscript I think this needs to be addressed. A few things that could be considered: The reader has no idea how important an increase of 0.005 km-1 extinction is (the value for the bulk of elevated smoke/polluted dust EC in Figure 2a). Is this an important magnitude? Could this amount of extinction result in the magnitude of T (2K throughout troposphere) seen in Figure 5?

Before explaining further, it is important to note that we are now stratifying our datasets by surface RH instead of surface pressure in addition to DM, as surface RH provides a more thorough filtration of high fire conditions.

As per the revision:

"The fingerprints of a traditionally-defined semi-direct effect where aerosols coincide with clouds would entail an anomalous warming of the cloud layer, and a corresponding decrease in RH. However, the meteorological conditions around which fires tend to occur need to be considered. As previously stated, large fires tend to occur during fire weather, which includes hot, dry, and windy conditions (Varga, 2022). Hot and dry conditions themselves are associated with high pressure anomalies in this region (Figure S4). Therefore, these fire weather conditions need to be "filtered out" to determine any potential semi-direct effects. Therefore, in addition to DM, variables need to be stratified by a second variable to account for the influence of meteorology on P, CF, and cloud properties. Fire season data was stratified by high (75th percentile) vs low (25th percentile) Ts, RHs, Us, and surface pressure to determine which variable was associated with the largest DM, and successfully filtered out fire weather condition anomalies... RH s was chosen as the second stratification variable, as stratifying nCA DM by high (RHs75) and low RH s conditions (RHs25) and differentiating the means of these distributions yields a significant DM anomaly [that] is an order of magnitude higher than the differences in mean DM between high and low conditions of the other potential stratification variables (surface pressure, Ts, and Us) (Figure S4, Figure S5, and Figure S4). This indicates that fire occurrence/fire emission are more dependent on RH s than these other fire weather variables."

We now stratify by 75th/25th percentile surface RH (as opposed to 90/10) to include more data into our distributions. This does change the averages and spreads of the distributions, but we feel that this new stratification removes much more variability than the previous method as evident with the larger shifts between the means of the 75th and 25th percentile distributions of each fire weather variable when stratified by surface RH. For example, tropospheric RH is 14.17+/0 0.73% lower under low surface RH conditions compared to high surface RH conditions (Figure 2e). This is around 7% higher than the difference in the means of the stratified tropospheric RH by surface pressure (Figure S4e).

We have replaced the CALIPSO profiles in the main text with MERRA-2 black carbon mass mixing ratio profiles, which are more spatially, and temporally, consistent with Aqua data. Therefore, we are no longer analyzing the data in terms of aerosol extinction. However, we have included analysis of CERES radiative profile. Specifically, we have calculated the heating rate due to the radiative contributions of aerosols (all-sky minus no aerosol profiles). We find that between 850-500 hPa, there is a heating rate of about 0.04 Kday-1 on (DM90/RHs75) days compared to (DM10/RHs75) days, and 0.09 K day-1 under (DM90/RHs75) compared to (DM90/RHs25) days (Figure 5). A prior idealized (10xBC) multi-modeling study found a global mean annual mean BC atmospheric shortwave heating rate of about 0.08 K day-1 was associated with around 0.8 K of warming in a model (Stjern et al., 2017). Assuming linearity, the 0.04-0.09 Kday-1 observed in Figure 5 would therefore correspond to around a 0.4-0.9 K temperature anomaly. Therefore, these heating rates can reasonably explain part of the approximately 1 K temperature anomaly found through the 850-500 hPA pressure levels.

Another study by Thornhill et al. (2018) conducted over South America during high vs low fire conditions found a heating rate of up to 0.2 K day-1 in the low-to-mid troposphere, and they record a similar, but stronger, reduction in clouds and precipitation as our study. Therefore, it is very feasible to assume that a smaller heating rate such as the one found in our study could create a similar, but weaker, semi-direct effect. Additionally, we find that the significant positive temperature anomaly in our study is no longer present at or above 250hPa, which is the same level in which the BC mass mixing ratio becomes insignificant.

Also, where are the clouds? Do you see the reductions in cloud fraction in the same place as the enhanced T?

See the earlier comment: "Additionally, we have added AIRS cloud fraction profiles. These profiles show a decrease in clouds coincident with increases in T as well as decreases in RH from 400hPa-600hPa. According to MERRA-2, there is also a significant increase in black carbon at these levels". We have also included five more figures that show the spatial distribution

of cloud fraction, temperature, relative humidity, water mass mixing ratio, and black carbon. Yes, spatially we generally see concurrent increases in temperature, BC, and corresponding decreases in CF, RH. This is true for all cells where clouds decrease except for a few grid cells in northeastern Nevada, but we provide a hypothesis for the reason for the cloud fraction anomaly in that area.

Finally, CALIPSO provides profiles on much shorter timescale and spatial scales. It may be beyond the scope of the paper to do the whole analysis, but it would be good to use a single year or a few cases to really establish that the increase in T, decrease in CF etc are definitively collocated with absorbing aerosol.

**This has been addressed in the above comments. We now use MERRA-2 BC profiles to obtain aerosol profiles that are spatially and temporally consistent with our other datasets.**

Extreme fire events are associated with the release of BB aerosols but also heat. Most, if not all, of the findings in this study could be explained by the coincident release of heat that increase the temperature, reduce the RH, and reduce cloudiness. How do the authors know that this isn't the case?

This is a reasonable hypothesis that is beyond the scope of this paper to determine. Therefore, it is listed as a possible contribution to the positive T/negative RH anomalies in the discussion with a proper reference. However, as stated earlier, we have demonstrated that at least part of the heating is due to shortwave absorption by BC associated with the fires. Therefore, we can state that the aerosol semi-direct effect at least partially contributes to this signal. Regardless, the title and final conclusions in the discussion are toned down.

The semi-direct effect (the main theme of the paper) is not introduced until Section 4. This needs to be properly introduced.

**Thank you for pointing out this oversight. The semi-direct effect is now properly introduced in the introduction.**

Focus on high/low extreme surface pressure conditions. I think it is important to stratify the large-scale environment etc to isolate on the aerosol impact and the authors have done a good job attempting this. However, the bulk of the results persist with a focus on contrasting the extreme (10th/90th percentile) conditions. The conclusions are dependent on the high or low ps conditions, but we don't know which conditions are more appropriate. Would it be useful to use the most common conditions so that it is more readily applicable to the real world? Or conditions that are more representative of future scenarios for the region of interest? Or at least include this element alongside the extreme conditions.

The stratification for surface RH (formerly surface pressure) has been changed from 90th/10th percentile to 75/25th percentile. This increases the number of days included in each distribution, therefore making them more reflective of everyday conditions. The purpose of the stratification is not necessarily to project how fires affect weather during most conditions, but rather to remove fire weather conditions from our analysis to better isolate the effects of the fires on weather. Figure 2 makes it clear that fires generally occur on days of lower RH, and therefore these days would have a higher probability of having higher fire emissions. As far as why we want to continue to stratify the data by 90th/10th percentile fire emissions, the goal of this paper is to analyze the effects of large fire events on climate, not the effects of fires in general. This makes it necessary to stratify by extremes in this case. This has been made more apparent in the manuscript. Additionally, it has been noted in the discussion that the present's 90th percentile fire emission days will become more common by the end of the 21st century.

The aerosol-cloud interactions section (4.5) needs more attention. The conclusions are that ARI dominates over ACI and this is due to the semi-direct effect. The increase in IWP coincident with T is highlighted as a 'fingerprint' of a dominant radiative effect. Many studies have found that IWP may increase due to either ARI or ACI. This includes convective invigoration and other mechanisms. One of the primary mechanisms by which aerosols impact clouds via ACI is the suppression of the warm-rain process. What does the precip response show? Is this consistent?

Ice water path change in the new stratification method is an insignificant increase. Convective invigoration could possibly contribute to an increase in IWP in southeastern Nevada, where there is a single grid cell with a negative  $\Omega$  anomaly at 300hPa (Figure S16), a positive water mass mixing ratio anomaly at 200hPa (Figure 9) and a positive IWP anomaly (Figure S13). However, most specific humidity anomalies throughout the troposphere are not significant, and the positive anomalies are limited to below the high cloud layer. Either way, the sentence describing the increase in IWP is removed, as the change is not consistently significant over the region.

The precipitation response is not generally consistent with significant decreases in effective radius in either high or low surface RH stratification (Figure 13g, Figure S12, Figure S13, Figure S14). Therefore, the suppression of precipitation does not appear to be related to microphysical effects. This has been noted in the manuscript.

Does the cloud-field distribution change? Shallow cumulus may respond in a different manner than the deeper clouds.

In the new Figure 11, there is an increase in AIRS CF at 850hPa at the coast, indicating an increase in shallow cloud fraction. There is an increase in hydrophilic BC and increase in moisture in this region. The increase in moisture may be due to water vapor emissions from fires, convective updrafts from the fire which bring moisture into the troposphere, or potentially from

moisture advection from another region due to changing wind patterns. Further study is required to determine the specific mechanism.

The discussion section is missing any comparison with other studies and reads as an afterthought. As a reader I have no idea if this is complimentary to other studies or paradigm shifting. I am surprised this has been overlooked.

A section has been added to the discussion that compares the findings of this paper to other studies both inside and outside the region of interest.

L15: Frequency and severity of wildfires. Please include some values or statistics.

The trend in frequency and size (area burned) have been added to this section. Fire count has roughly tripled in the 1925-2019 timeframe and burned area has roughly quadrupled (Li & Banerjee, 2021).

L31: "..altering the hydrological and radiative balance of the atmosphere." Please include references.

References have been added.

L34. I don't think a reference to methane fits within this manuscript.

This line has been removed.

L36: "..leading to increased SW forcing". Which direction is the forcing?

This line has been clarified to indicate that there is a reduction in outgoing shortwave flux.

L37: "higher injection". Higher in altitude or magnitude?

This has been clarified. We are referring to the altitude of the aerosol.

L37: Can you expand this to include other aerosol impacts on cloud microphysics? Such as LWP, the warm-rain process, convective invigoration?

The introduction has been expanded to include aerosol-cloud microphysical interactions. Effects of aerosols on LWP, convective invigoration, and precipitation are now included.

L43: What other parts of the world? Please give examples of other locations that may experience similar events.

Citations have been provided that detail changes in the fire regime in the Mediterranean basin, Australia, and South America.

L44: "aerosols primarily and secondarily emitted". These terms need defining, especially 'secondarily'.

Primary emissions have been defined as those emitted through biomass burning, while secondary emissions have been defined as oxidation of emitted volatile organic compounds emitted from burning. We have also indicated wildfire feedbacks as a source of dust.

L45: What are BB emissions dependent on? What are these models missing? What is a 'proper' parameterization of BB emission (L54)?

Biomass burning aerosol emissions are dependent on modeled fire carbon emissions, which themselves are a function of flammability and plant functional type. Flammability itself is a function of relative humidity, soil moisture, and temperature in model fire parameterizations. Most CMIP6 models do not have interactive fire models, and of those that do, only a handful are capable of modeling vegetation dynamically, and fewer have parameterization of biomass burning aerosol emissions. This information has been added to the manuscript. By "proper" parameterization we meant interactive as opposed to prescribed emissions.

L48: Are there any studies that are relevant for the region you are looking at? This section should provide us with a good base for establishing where our current level of understanding is.

Further discussion on Twohy et al., 2021 as well as a climate modeling study conducted in 2014 over the region concerning wildfire aerosol forcing (Chen et al., 2014) has been included in the introduction.

L54: "dust aerosol from wildfire-cleared vegetation" needs to be properly introduced. Have other studies shown this process is an important source of uncertainty?

The literature on fire-dust feedbacks has been moved from L215 to the introduction.

L67: What is fire dry matter emission? Is this expressed in GFED as fluxes of different aerosol species? Can you include this information in the manuscript and properly define DM.

Fire dry matter emissions are emissions (gasses and/or aerosols) associated with the burning of dry vegetation. This has been specified in the revision.

L71: 'burned area' and 'fire power' need to be defined. Where does the fire power dataset come from? What is the difference between the two methods? If the burned area is from AQUA/TERRA does it not also suffer from cloud cover obstruction?

Burned area has been defined in the introduction. The MODIS burned area retrieval algorithm is now briefly described in Section 2.1, along with the fire power dataset retrieval algorithm. The fire power dataset is from MYD14A1. This has been cited in the revision. The burned area does suffer from cloud obstruction, but unlike fire power, burned area can be recorded on the next satellite overpass once the cloud has cleared. This does create a temporal uncertainty, but since the burned area dataset uses both Terra and Aqua satellites, this reduces the chances of a temporal uncertainty greater than one day.

L81: Why is there a temporal uncertainty? Is this related to the method for calculating burned area?

This has been addressed in the revision Please see the previous comment for the reason for the temporal uncertainty.

L86: "..GFED is from an older model". Older model compared to what? What has changed? Is it likely to impact the conclusions of this study? Please state which version of GFED this study uses in Section 2.

This section has been rephrased to highlight the uncertainties of the CASA model instead of simply calling it an old model. Specifically, the lack of meteorological terms in the calculation of net primary production is now highlighted.

L87: suggestion: replace accurate with robust.

The reviewer's suggestion has been adopted and accurate has been replaced with robust.

L100: Are the retrievals instantaneous? If so what local time are the retrievals valid for?

The Aqua satellite makes two overpasses for the region of interest: one ascending run from 2-3 PM, and one descending run from 2-3 AM. The descending (AM) dataset is used for MODIS and AIRS variables, as the ascending (PM) dataset is missing cloud property and AOD data.

L116: The SSF1Deg-Day product provides a daily mean value. Why was this used instead of the instantaneous SSF1Deg-Hr product? The latter is temporally collocated with MODIS and AIRS products on AQUA, whereas I believe the daily product will include interpolation of (AQUA) values across a synthetic diurnal cycle.

Thank you for bringing this to our attention. However, we have switched the CERES dataset we are using to the SYN1Deg dataset, as we have now included analyses of the atmospheric radiative profiles of the region. This product is produced through combining Terra and Aqua observations of radiative flux and cloud cover with GEOS estimates of temperature and relative humidity. We understand that this is not ideal for temporal consistency, but this sacrifice is necessary to estimate radiative heating of the biomass burning aerosols.

L137: Is it reasonable to assume all the variables are well-fit assuming a normal distribution? Instead of fitting a normal distribution, the empirical cumulative distribution functions are plotted. We feel that this gives a more real representation of the distribution, and we no longer have to worry about the data being well fit for a normal distribution.

L155: A lot of unnecessary detail in this section. Please consider reducing.

This section has been heavily revised to remove unnecessary detail..

L196: Is there a more scientific word that can be used in place of 'suspicion'?

This word has been replaced with "hypothesis."

L199: Would this be better placed in Section 3.2.

This text has been moved to section 3.2.

L215: Aha! Here is the fire-dust introduction. I suggest moving this to the introduction section.

We have adopted the reviewer's suggestion. The literature on fire-dust feedbacks has been moved from L215 to the introduction.

L218: 'Semi-direct effects'. This term hasn't been introduced. Given that this is a theme of the paper and is in the title I think it should be given a proper introduction.

Aerosol semi-direct effects are now defined in the introduction.

L221: "dust and smokey aerosols". Could you briefly explain how CALIPSO classifies this species?

CALIPSO is no longer utilized in the main text. However, an explanation of this has been added to the CALIPSO section of the supplement. Essentially, polluted dust is classified as a combination of wind-blown dust from the AERONET coarse dust clusters and fine biomass burning aerosols clusters.

L224: How important is an extinction of 0.01 to 0.02 km-1 of extinction? How does this compare to background extinction profiles?

This is addressed in our new analysis. Instead of using extinction profiles, we are using CERES radiative flux profiles. We find that there is a roughly 0.04 K day-1 heating rate between 850-500 hPa in the troposphere due to the biomass burning aerosols during high surface RH days, and a 0.09 K day-1 heating rate for low surface RH days.

L229: Negative anomalies. This requires more attention. I don't think it is sufficient to point towards large error bars – especially given they are deemed statistically significant. If you suspect an abnormal month then it should be possible to check this.

These large negative anomalies are no longer present with the alteration of the algorithm used to generate the CALIPSO profiles. The new algorithm compares monthly anomalies as opposed to just taking a difference between monthly averages. Additionally, CALIPSO is no longer utilized in the main paper.

L234: "fingerprints of SDE = warm cloud layer". This example is one of many ways the SDE may manifest, and only holds true if the absorbing aerosol is located within the cloud layer. Please rewrite to make this clearer.

This sentence has been rewritten to be clearer.

L236: CDF is already defined.

This oversight has been corrected.

L243: "distribution of AOD is significantly different". Are you referring to the spatial distribution? If so where do you show this is significant?

This line has been removed because of the new results.

L267: MH20 is not defined in the main manuscript. Is this from AIRS?

This was defined in the manuscript, "Aside from temperature, the other potential factor that could affect RH is that of specific humidity, which is analogous to water mass mixing ratio MH2O." This is from AIRS and is now defined earlier in the satellite data section.

L270/275: what is the mechanism driving low specific humidity in the high troposphere? Could you summarise the findings that are in section 2 of the supplement?

While determining this mechanism requires further study beyond the scope of this paper, there is a positive wind speed anomaly in the cloud layer of that region that leads towards a positive water mass mixing ratio in CA and southeastern/ Nevada. This implies that moisture advection is transporting moisture from northeastern Nevada to elsewhere. Further study is required to determine if this is the case, and if the wind speed anomaly in that region is related to the BB aerosol forcing, or if it is just an artifact that exists in the stratified dataset.

L282-283: This information is in the figure caption and doesn't need to be re-introduced.

The manuscript has been edited to reduce re-introductions.

L284 (+288 +290): "Significantly leftward". I don't think a direction is a useful metric. Also, I'm not sure the figure shows CF liquid for ps90 under high DM is significantly suppressed (leftwards).

Wording has been fixed to be clearer. Now this shift is described as a "shift towards a higher probability of negative anomalies."

L286: "(at the 90% confidence interval)". This has already been defined.

The manuscript has been edited to reduce re-introductions.

L287-288: The difference in CTH anomaly is significantly increased for ps90.

Wording has been fixed to be clearer.

L293: "this creates conditions of cloud-free skies". Do you observe largely cloud-free skies under ps90 DM10 conditions?

This is shown in Figure 3f of the preprint (now Figure 2f). Under low surface pressure (or in the new submission, high surface RH), cloud fraction anomaly is significantly more likely to be strongly negative. Apologies if this wasn't made clear in the paper, it has been edited to point this out more specifically.

L321: Can you suggest a mechanism that explains the change in cirrus cloud fraction?

There is a positive wind speed anomaly in northeastern NV that may be advecting moisture outside of that region.

L329: "IWP scales positively with T, so this is a fingerprint of a dominate radiative effect". How do you know this is not confounded by microphysical effects? Could IWP responses not be driven by ACI and temperature via ARI?

Changes in IWP are not significant and are weaker under the new stratification. This section has been removed.

L334: There is a lot of observational evidence showing biomass burning aerosols can enhance cloud droplet number concentrations. Please include these studies as a contrast.

Studies showing that biomass burning aerosols can enhance cloud droplet number concentrations have been included, particularly in an overlapping region

L338: "...reduce RH to the point where clouds are unable to form in the first place". Can you see signs of this in the dataset you have? A simple histogram of LWP or CF would give you an idea of how the cloud field distribution is modified.

The section was worded poorly. What we meant was that as relative humidity is lower under these conditions, the probability the skies will be cloud free, or have a negative cloud anomaly are high. This is illustrated in Figure 2e. Therefore, there are fewer clouds available to burn off. The text has been edited to make this clearer.

L346: Longwave effects. Is there a reason this analysis wasn't extended to look at the LW or NET radiative fluxes at TOA available in the SSF1deg product? Analysis of net TOA forcing due to the reduction of clouds is now included. Additionally, upward as well as downward LW radiative profiles are included in the supplement.

L355: Results may be applicable to other Med climates. Please expand on this. Do they have similar atmospheric drivers?

**This line has been deleted.**

L363: "few and far between / infancy". There have been advances since 2016 to include interactive fire emissions. I suggest the authors include more recent studies.

Two newer studies have been added to this reference, and the wording has been changed to convey that these fire modules are not widely used in modeling studies.

All figures: Once something is defined (i.e., Northern California nCA, 10th percentile DM DM10) I don't think you need to define it again in every subsequent figure. This would help clarify the figure captions. This is also true for the manuscript (i.e., pressure ps).

The manuscript has been edited to reduce re-introductions.

All figures: Not always easy to read the labels and axis labels. Scrolling through the pdf at 100% zoom will show you the plots that need amending.

**Figure text has been resized.**

All map plots: I suggest replotting the plots showing the spatial distributions so that the latitude and longitude cells are equivalent. In all plots the latitude is stretched. I also suggest making the 'significance dot' larger so it's immediately clear.

The plots have been made to have equal axes, and the size of the significance dot has been increased.

Figure 5. Label for M\_H2O includes a subscript (cl). What does this indicate?

This was a typo; the variable is now properly labeled.

Figure 7: Caption details for (a) and (b) are switched.

This typo has been corrected.

Figure 9: Colorbar labels disappear at bottom of range.

This figure is now properly sized.

Figure 11. LWP and IWP wrong units shown.

The correct units of g m-2 are now shown.

Reviewer #2:

In their paper, Gomez and coauthors stratify observations of meteorological parameters over northern California by surface pressure (as a proxy for fire weather) and fire emissions and find that for both pressure extremes analyzed, greater biomass burning emissions correspond to warmer temperatures, lower relative humidity, and lessened cloud fractions. They interpret this as evidence for a semi-direct effect that reduces cloud fraction from smoke absorption.

Unfortunately, no evidence is presented that clearly demonstrates this relationship is both causal and that it runs in the direction the authors assume. In particular, a reasonable hypothesis based on the same results would be that, for the same general synoptic conditions, random fluctuations that increase temperature (decrease RH) favor increased fire emissions.

New evidence has been added that demonstrates the link between aerosol absorption and T/RH changes: We have included analysis of CERES radiative profiles. Specifically, we have calculated the heating rate due to the radiative contributions of aerosols (all-sky minus no aerosol profiles). We find that between 500-850hPa, there is a heating rate of about 0.04 Kday-1 on (DM90/RHs75) compared to (DM10/RHs75), and 0.09 K day-1 under (DM90/RHs75) compared to (DM90/RHs25) days (Figure 5). A prior idealized (10xBC) multi-modeling study found a global mean annual mean BC atmospheric shortwave heating rate of about 0.08 K day-1 was associated with around 0.8 K of warming in a model (Stjern et al., 2017). Assuming linearity, the 0.04-0.09 Kday-1 observed in Figure 5 would therefore correspond to around a 0.4-0.9 K temperature anomaly.

Another study by Thornhill et al. (2018) conducted over South America during high vs low fire conditions found a heating rate of up to 0.2 K day-1 in the low-to-mid troposphere, and they record a similar, but stronger, reduction in clouds and precipitation as our study. Therefore, it is very feasible to assume that a smaller heating rate such as the one found in our study could create a similar semi-direct effect. Additionally, we find that the significant positive temperature anomaly is no longer present at or above 250hPa, which is the same level in which the BC mass mixing ratio becomes insignificant.

The hypothesis that random fluctuations in temperature and RH has been addressed through the new stratification method. Instead of using surface pressure, we now use surface RH. We also now stratify by 75th/25th percentiles instead of 90th/10th percentiles for surface RH to ensure the stratification contains as much data as possible without compromising the filtering out of any weather conditions that contribute to enhanced fire activity. Comparing Figure 2 to Figure S4, where we show the surface pressure stratification provides much weaker stratification of fire emissions and fire weather conditions compared to surface RH for the new 75th/25th percentile stratification. In fact, DM under low surface RH conditions compared to high surface RH conditions (Figure 2a) compared to DM under low surface pressure conditions (Figure S4a). Ultimately, the point of this stratification is to ensure that when we are comparing high fire conditions to low fire conditions, that the weather conditions surrounding both scenarios are as similar as possible, and that the differences are due to the fire itself, and not due to any weather conditions around the fire.

The paper is not suitable for publication as written. One option to move forward would be to conduct some kind of analysis that would plausibly establish causality. For instance, are the temperature perturbations seen very large compared to variability within a given synoptic state? (And thus require an explanation beyond meteorological variability.) The other option, which is more feasible if the authors are to limit themselves to an observational analysis, would be to reframe the results as an interesting hypothesis versus a shown conclusion. The work would then argue for follow-up analysis using a model (perhaps a regional model given GCM limitations) that could assess the plausibility of effects of this size from a given smoke perturbation. This wouldn't require much additional analysis, but would require a major rewrite of the title, abstract, and main text.

As per the above comment, we have performed extensive analysis to show that the biomass burning aerosols are associated with a positive heating rate that contributes to the temperature anomaly. The title has been edited to tone down the attribution of the temperature anomaly to the semi-direct effect of these aerosols, and in the discussion, we note that there may be other contributing factors to this anomaly, such as sensible heat flux from the fires themselves.

Line 129: CALIPSO does have snapshots available; it's fine if you don't want to use them, but maybe specify that you're just referring to gridded products here?.

CALIPSO is no longer utilized in the main text. Instead we now use MERRA-2 for our BC profile, which provides data that is spatially and temporally consistent with the other datasets.

Figure 5: The bars currently show one standard error, correct? They should be changed to 90% confidence if you want to use "significant" language.

All error bars are now 90% confidence intervals.

Figure 7: Panels a/b are mixed up in the caption versus the figure.

This typo has been corrected.

Code availability: "Upon request" is not compliant with ACP standards, in my reading. I would advise creating a Github repository.

A github repository has been created, and MATLAB functions created to process the data will be uploaded once the paper has finished the review process.

Independent of the reviewers comments, we would like to indicate that we have switched from using the MODIS "Deep\_Blue\_Aerosol\_Optical\_Depth\_Land\_Mean" product and instead use the "Aerosol\_Optical\_Depth\_Land\_Ocean\_Mean" product, as this is the product that the SYN1Deg uses to calculate their radiative flux profiles.

---

## Author Response (AR2)

We would like to thank the reviewer for their comments. In summary, in this new revision we have removed any references to the semi-direct effect being the main contributor to the anomalies depicted in the Figures. However, we still indicate that our analysis shows that the aerosols are contributing to the temperature anomalies, as the heating rate associated with the aerosols is significant and positive, and is not due to changes in water vapor. We note that to estimate the exact contribution of the smoke aerosols to the meteorological anomalies, a modeling study must be conducted. We have also changed the title to "California Wildfire Smoke Contributes to a Positive Temperature Anomaly over the Western United States" to reflect that the heating rate contributes to the positive temperature anomaly, but removes any reference to a semi-direct effect occurring.

In addition, a few sentences have been edited for clarity or accuracy.

The CALIPSO discussion is also confusing. CALIPSO data is fairly coincident temporally and spatially (albeit with a narrower footprint) with Aqua. That's the whole preimise of the A-train! The use of MERRA-2 for BC profiles may very well be an improvement, but representativeness in this sense is not plausibly the reason.

We apologize that this point was not made more clearly. The resolution of the gridded CALIPSO dataset is much too coarse to create profiles that are consistent with the AIRS generated profiles. Additionally, the gridded dataset is monthly. CALIPSO does not sample as large of an area as MODIS on the same overpasses, leading to missing data over wide regions on most days. This makes the MERRA-2 dataset more appealing, as it does not have missing data, and is available at finer timescales. Additionally, the MERRA-2 dataset is a finer resolution than MODIS, so it can be interpolated to 1 degree.

Additionally, CALIPSO started collecting data in 2006, while Aqua MODIS  and AIRS data goes back to 2002. Therefore, the CALIPSO satellite is not temporally consistent with AIRS/MODIS in that the data is not available for the same time period.

The heating rate discussion seems to assume that only DM could be contributing to the SW absorption, but water vapor could also both directly and via swelling of the aerosol particles and lensing effects.

We would like to thank the reviewer for this alternative interpretation of the dominant contributor to the heating rate. However, in the revision, this was already demonstrated to not be a possibility.

Below is Figure 6 from the revision. You may see that **regional average differences in specific humidity are insignificant throughout the troposphere, no matter the stratification (see panels b,f).** Therefore, this is not a valid explanation for the **significant** increase in aerosol shortwave heating. This has been made clearer in the revision. Therefore, the only plausible explanation is that the absorbing properties of the BC are leading to this

absorption and heating. This heating rate therefore also addresses the issue of causality that is brought up. We can say with certainty that these aerosols therefore **contribute** to the observed temperature anomalies, and therefore likely contribute to the decrease in RH, though we don't know to what extent the aerosols contribute to these anomalies.

[Figure]

Furthermore, looking at a spatial scale, the heating rate from 850 hPa-500 hPa is spatially consistent with the widespread BC anomalies, not the scattered and scant significant water mass mixing ratio anomalies.

[Figure]

[Figure]

[Figure]

[Figure]

The authors "assume linearity" to go from a 0.04-0.09 Kday−1 heating rate to "around a 0.4-0.9 K temperature anomaly." I don't understand why they would try to do the analysis in this way. Shouldn't the heating rate itself be interpreted as the maximum change in temperature to be expected within one day due to shortwave absorption? I also have no idea why we would assume linearity holds for global-averaged values of lots of different regimes for BC perturbations versus the regional effects here.

This is a fair point, and we would like to thank the reviewer for this comment. We have removed this line. However we would like to reiterate that prior work indicates that our observed heating

rate is consistent with another study that is regionally based. Thornhill et al., 2018 found that absorbing aerosols in a model, prescribed from aircraft observations, were associated with a maximum heating rate of 0.2 Kday-1, which is roughly 2x-5x that of the aerosol heating rate that we determined, which is consistent with their higher concentration of BC. They also witnessed a decrease in CF that was a similar magnitude to what we saw in our results, with significant negative CF anomalies of 0.08.

Are the authors sure that, within their RH quartiles, within-quartile variations in RH or other parameters aren't driving any of the observed meteorological differences between the high and low DM deciles? I don't see any analysis to rule this out. It would be difficult in any event, which is why it's hard to infer causality from observations in the first place!

The authors state that binning by the quantiles as they've done means the effect of "fires can be discerned independent of the meteorological conditions that come with high DM extremes", but this is simply not true in a rigorous sense. Pretty much all of my original objections still apply to the revised work. I therefore cannot recommend publication barring major revisions in analysis or framing.

This is a fair point, and we will include the caveat that causality is difficult to discern using this method. However, there is evidence that the fires are contributing to the T anomalies outside of the anomalies present in this stratification alone.

Firstly, there is the aforementioned aerosol heating rate. The fact that this heating rate is significant and positive, and we know it is not specific humidity driven, indicates that the aerosols are highly likely to be contributing to the temperature anomalies, at least from 850 hPa-500 hPa.

Additionally, in every spatial map, we have an r-value that depicts the cross correlation between AOD and the variable of interest. These cross correlations are generally statistically significant and large in magnitude, especially for cloud layer temperature, indicating that there is a relationship between intensified AOD and these anomalies in each gridcell.

[Figure]

Furthermore, the temperature spatial distribution, both horizontally and vertically, of the temperature anomalies are consistent with the location of the BC. See Figures 7 and 8 below, which depict BC and T and multiple vertical levels under the same conditions as the figures above. The spatial distribution of temperature anomalies and BC anomalies in CA and southern/eastern Nevada are spatially consistent, and the positive temperature anomalies generally become weaker and/or disappear at the same pressure levels that the positive BC anomalies become weaker and/or disappear.

[Figure]

[Figure]

[Figure]

Due to all the above presented evidence, especially the positive aerosol heating rate, we are confident that the black carbon aerosols contribute to the temperature anomalies at the very least. We have included the caveat that while the aerosols are contributing to the temperature anomaly, we do not know to what extent they are, and by extent we do not know how they are affecting the RH or CF anomalies. We note in the discussion that there are other possible contributors to these anomalies, such as sensible heat release from the fires, the circulation anomaly over northeastern Nevada, or random weather fluctuations. We also now include the caveat that the stratification analysis does not completely remove the issue of causality. However due to the aforementioned heating rate of the aerosols, as well as the spatial consistency between the location of the aerosols and the T anomalies from 925 hPa-500 hPa, we are confident that the aerosols at least contribute to the temperature anomalies that are

observed. We indicate that modeling is needed to determine the exact amount to which the smoke contributes to the increase in temperature, and decreases in CF/RH.